



# 60-years analysis of drought in the western Po river Basin

Emanuele Mombrini[1], Stefania Tamea[1], Alberto Viglione[1], and Roberto Revelli[1,†]

[1]Dipartimento di Ingegneria dell'Ambiente, del Territorio e delle Infrastrutture (DIATI), Politecnico di Torino, Torino.
[†]deceased, 6 May 2023

**Correspondence:** Emanuele Mombrini (emanuele.mombrini@polito.it)

**Abstract.** Since the start of the 21st century, greater focus has been put on drought and its wide range of environmental and socioeconomic effects, particularly in the context of climate change. This is especially true for the North-western region of Italy, comprising the Piedmont and Aosta valley, which have been affected in recent years by droughts that have had acute effects on water resources and water security in all sectors, including agriculture, energy and domestic use. The region also belongs to the Mediterranean hot-spot, characterized by faster than global average warming rates and higher vulnerability to their effects. Therefore, characterizing the observed changes and trends in drought conditions is of particular significance. To this end, 60 years of precipitation and temperature data from the North West Italy Optimum Interpolation data set are used to calculate the drought indices SPI (Standardized Precipitation Index) and SPEI (Standardized Precipitation Evapotranspiration Index) at a shorter (3-month) and at a longer (12-month) time scale. First, trend analysis on precipitation and temperature is performed, finding limited areas with significant precipitation decrease and, conversely, a general temperature increase over the region, with higher values found in the higher elevation areas. Changes in meteorological drought are then evaluated, both in terms of drought indices trends and in terms of changes in the characteristics of drought periods, on both a local and regional scale. A relation between the altitude of the area and the observed changes is highlighted, with significant differences between the plain and mountainous portion of the region. The differences are mainly related to the observed trends, with the low-altitude part of the region displaying a tendency towards dryer conditions not in common with the mountainous area. Significantly, no trend is found at a region-wide level but is instead found when considering homogeneous areas defined by terrain ruggedness. Furthermore, changes in the number of drought episodes and in their severity, duration and intensity are found to be correlated with terrain ruggedness at all time scales.

## 1 Introduction

Drought is considered to be one of the main natural disasters, with severe and widespread effects. Drought has caused, in the period 1979-2019, 650000 deaths worldwide (World Meteorological Organization, 2014), which rise to almost 12 million when considering the 1900-2023 period (EM-DAT and CRED / UCLouvain, 2023). Despite accounting for just 5% of total natural disasters, more than one billion people have been affected by it between 1994 and 2013 (Wallemacq et al., 2015). Drought is also linked to severe financial costs, as it has led to losses in the range of 332 billions of USD since 1980 in the US alone, with average losses for single drought events in different countries ranging from 0.03% to as high as 1.6% of the GDP (García-León et al., 2021). Ecosystems greatly suffer from drought, with vegetation water stress, wildfires, widespread animal mortality and



hydrological ecosystem collapse becoming increasingly evident (Crausbay et al., 2020). Drought also has both short and long terms effects on water availability (IDMP, 2022), and they are relevant when considering the global increase in water demand in the last 100 years and the predicted challenges in meeting that demand in the future (Unesco, 2018; Wada et al., 2016; Burek

et al., 2016).

These drought-related phenomena are also likely to become more intense, as droughts are predicted to become more severe and frequent under climate change conditions (Dai, 2011, 2013; Trenberth et al., 2014; Ward et al., 2020; Pörtner et al., 2022). Understanding how and where changes will occur on a local scale is necessary in order to develop adequate responses. Several studies on drought trends in the northern Italian peninsula—often in the context of the wider Mediterranean or alpine region—

have been carried out, analyzing precipitation and temperature series through the use of drought indices in order to understand changing patterns of meteorological drought. In general, an increase in drought occurrence in the area has been detected, even if analysis on longer time scales (Haslinger and Blöschl, 2017; Hanel et al., 2018) have not considered recent drought events as exceptional when compared with historical records. Causes and dynamics of the reported changes differ significantly when based only on precipitation data, with precipitation decrease found either in the winter or summer season (Bordi and Sutera,

2002; Brunetti et al., 2002; Hoerling et al., 2012; Haslinger and Blöschl, 2017; Pavan et al., 2019). On the other hand, studies considering both precipitation and temperature (Hanel et al., 2018; Falzoi et al., 2019; Arpa Piemonte and Regione Piemonte, 2020b; Vogel et al., 2021; Baronetti et al., 2022) have found more consistent results, indicating the rise in evapotranspiration as a main factor in drought increase, even when significant changes in precipitation patterns were detected.

In addition to this focus on drought trends in wider areas, interest in regional expressions of climate change has been

growing. One of the most investigated of these regional phenomena is the enhancement of warming rates with elevation, known as elevation dependent warming (EDW), on which many studies have been performed in recent decades (for a summary of studies based on surface measurement, see Mountain Research Initiative EDW Working Group, 2015, for a summary of studies based on climate models, see Palazzi et al., 2019). In general, despite sometimes conflicting results and the lack of adequate climate data for mountainous regions, a consensus on enhanced warming rates at higher altitudes emerges (Rangwala

and Miller, 2012; Pepin et al., 2022). Orographic precipitation gradients, meaning the elevation-dependency of precipitation change (EDPC), while less understood, have also been widely investigated. Finding a consensus is more difficult in this case: the comprehensive meta-analysis of Pepin et al. (2022) reported a relative decrease in precipitation compared to lowlands, although without high confidence; furthermore, analyses such as Giorgi et al. (2016) have shown the importance of spatial resolution in understanding these processes, reporting opposite results at high/low resolutions. Given the particular orography

of the northern part of Italy, defined by the wide Po plain and the surrounding alpine chain, possible links between observed drought changes and terrain characteristics are therefore also of interest, as they could lead to different and even opposite trends in sub-regional areas.

Thus, the aim of this study is to investigate if the estimated changes in drought conditions found for north Italy also apply to the smaller Piedmont and Aosta valley region in north-west Italy, and how the estimated increase in drought conditions

may differ between smaller zones. In particular, possible correlations between orography—meaning both elevation and terrain

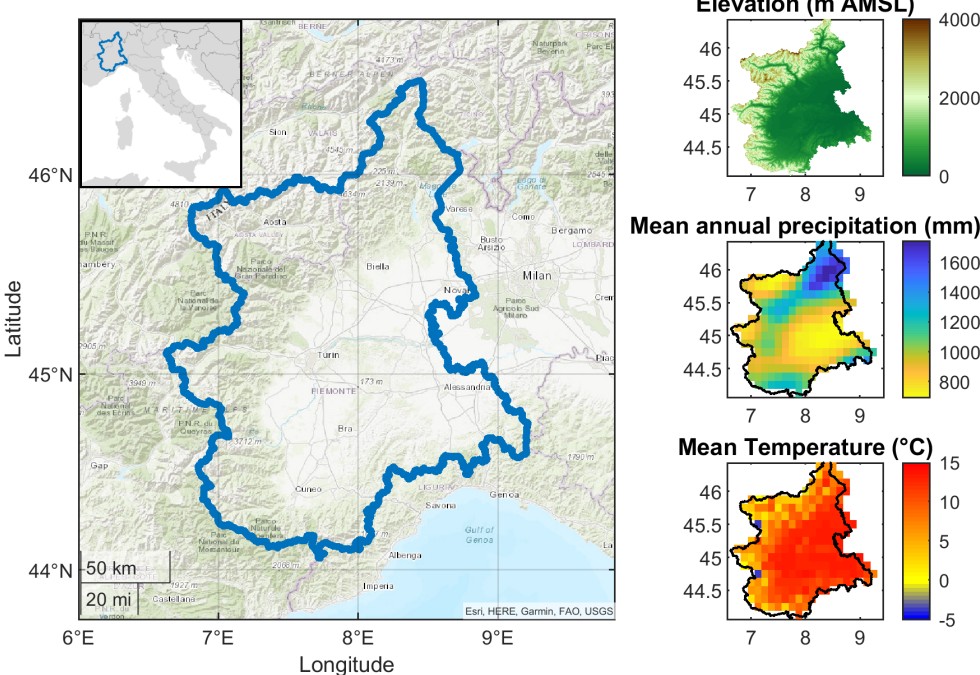

**Figure 1.** Map of the study area, including elevation, mean annual precipitation and mean temperature values.

ruggedness—and drought are evaluated, both in terms of the distribution of wetting/drying trends, and in terms of different dynamics (duration, severity, intensity) of drought periods. To this end, a fine-grid data set of precipitation and temperature values spanning more than 60 years is analyzed by calculating the SPI and SPEI at a shorter 3 month time scale and at a longer 12 month time scale. The index series were analyzed in order to find trends in drought conditions, as well as changes in drought characteristics both on a local and on a region-wide level.

In Section 2 the study area, as well as the analyzed data set are presented; the analysis methods used are also discussed, including the derived SPI and SPEI indices. In Section 3 the results obtained from each analysis are reported, while in Section 4 the general conclusion derived from the study are discussed.

## 2 Materials and methods

### 2.1 Study area

The Piedmont region and the Aosta valley (Figure 1) are situated in the north-west of Italy, bordered by France on the east and south-east, Switzerland on the north, and two other Italian regions (namely Lombardia and Liguria) on the west and south-west. Although divided in two regions on an administrative level, they are both part of the Po river basin, and together they comprise





the Italian north-western side of the alpine chain; furthermore, the main valley in the Aosta region leads directly into the
Piedmont territory. Therefore they are going to be treated as one domain for this study. The Piedmont part of the domain covers
more than 25000 km$^2$, while The Aosta Valley part covers 3200 km$^2$, the whole area having latitudes ranging from 44.06°
to 46.44° N and longitudes ranging from 6.69° to 9.19° E. Orographically, it is mainly a mountainous region, surrounded by
the Alpine chain from the south-east, throughout the western front, up to the norther point. Mountainous reliefs occupy half
of its territory, with the highest peaks lying in the Aosta Valley, while the Po plain lies in the central and eastern part of the
region. Due to the small latitude variation throughout the area, the area's reliefs plays a key role in defining the area's climate
variability (Arpa Piemonte, 2007): continental air masses from the Po plain, moist currents from the Mediterranean sea and
north-western Atlantic currents interact with the orography leading to a complex and spatially variable climate (Ciccarelli et al.,
2008). Precipitation is characterized by a bimodal distribution, with maxima in spring and autumn, and minima in summer and
winter. For most of the region (close to 90%) winter is the season with the absolute minimum precipitation, and only for the
south west corner the absolute minimum is in summer; for the western and southern side of the territory (close to 60% of the
overall area) the highest precipitations occur in autumn, while for the central-eastern part they occur in spring (Perosino and
Zaccara, 2006). Annual precipitation ranges from 700 to more than 1700 mm, with a mean of 1000 mm. Annual precipitation
is lowest in the central-west area and in the Aosta Valley, while highest in the norther area. Mean annual temperatures range
from slightly over 13°C near the eastern border to slightly under -3.6°C, closely following the altitude of the area; the diurnal
temperature range varies from 2 to more than 12 °C, is highest in summer, and is also generally inversely correlated with
height.

## 2.2 Data source and data processing

The data used in the analysis is obtained from the North Western Italy Optimal Interpolation (NWIOI) data set, calculated and
published by the Forecast Systems Department of the Regional Environmental Protection Agency of Piedmont (*Dipartimento
Sistemi Previsionali - Arpa Piemonte*). The data set contains daily precipitation, maximum and minimum temperature values
over a regular $24 \times 20$ grid, covering the area 6.5-9.5 W and 44.0-46.5 N with a 0.125° resolution and WGS84 projection. The
data is obtained through an analysis of the region's meteorological station network data via the Optimal Interpolation method.
The method spatially interpolates station data by correcting a previously defined background field; the correction are based on
the station data by defining an "area of influence" for each station. This area of influence was both horizontal and vertical in
the case of temperature stations (tri-dimensional interpolation) and only horizontal in the case of precipitation stations (two-
dimensional interpolation). In any case, no direct trend relation between the meteorological values and elevation has been
evaluated and removed/added to the data (further details on the Optimal Interpolation method and its calibration are given in
Appendix A).

Although the data is provided as already analyzed and validated, some errors are found: a small percentage (slightly under
1% ) of data reported the maximum daily temperature as lower than the minimum daily temperature. Given that the data set
is used to calculate monthly means, such data is discarded. For the purpose of the subsequent analysis, 227 (the number of

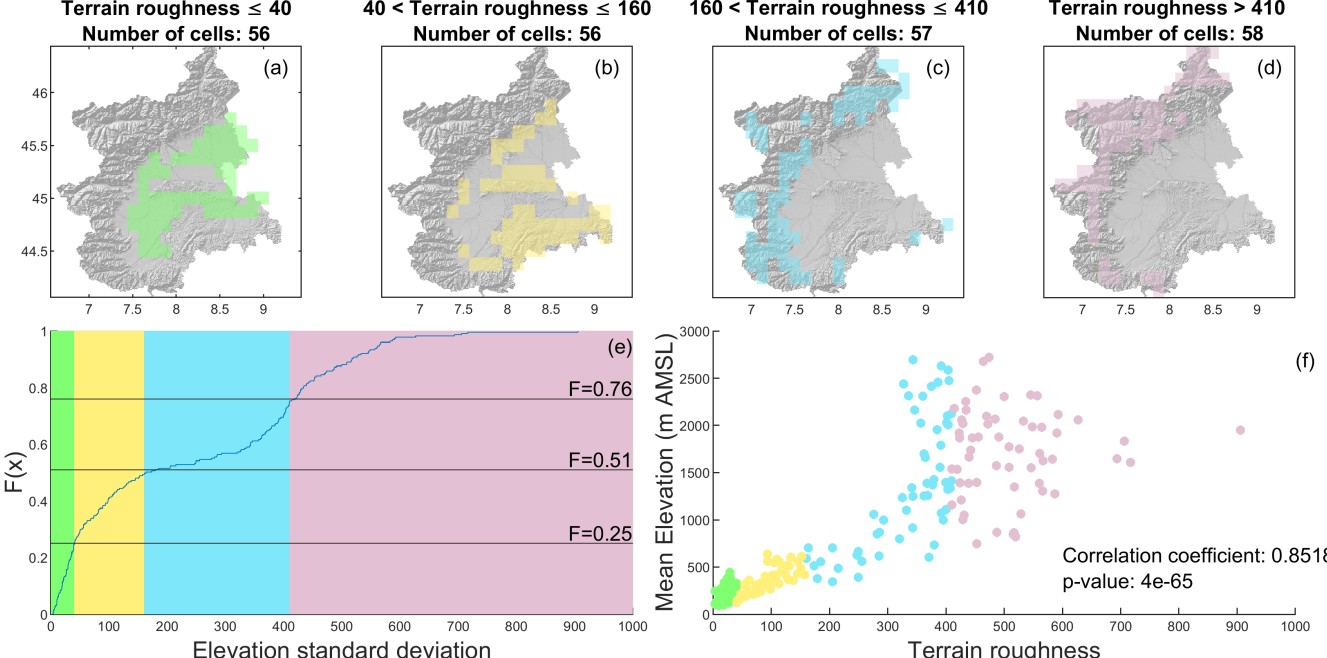

**Figure 2.** Areas defined based on terrain ruggedness, calculated as standard deviation of the elevation inside each cell and displayed in the first row. **(a-d)** Areas belonging to each class, corresponding to the A to D areas cited in the following figures. **(e)** Cumulative distribution of elevation standard deviation values. **(d)** Correlation between elevation and elevation standard deviation.

grid points inside the region) series of 783 monthly values (December 1957 - February 2023) of precipitation, maximum and minimum temperature are calculated.

## 2.3 Area definition based on elevation

Trends in precipitation, temperature and drought indices are also compared between four distinct areas, representing the plains, the hilly region, and the lower and higher mountains respectively (Figure 2). Due to the orography of the domain, these areas are not defined based on an elevation threshold; rather, the thresholds used are based on the ruggedness of the terrain inside every $0.125° \times 0.125°$ cell. By calculating the standard deviation of the height based on the EarthEnv-DEM90 digital elevation model (Robinson et al., 2014) with 90 m resolution, four areas with an almost equal number of cells are defined. Significantly,

the classification was able to distinguish the hills in the center-south of the region from the eastern flat part of the region, despite similar mean elevation. As stated in the previous section, orography, meaning the combination of elevation and terrain roughness, are an important factor in the area's climate and, as such, dividing areas with similar elevation but distinct terrain (such as flat and hilly areas) is considered important for the purposes of this study. Still, four areas based only on mean elevation thresholds (260, 600 and 1600 m AMSL) and similarly of an equal number of cells were also considered as a comparison to





the proposed classification, and the difference in the obtained results is described in 3.2. Furthermore, as will be discussed in Section 3.3, both terrain ruggedness and elevation were found to be significantly correlated with drought characteristics and their changes over time but, crucially, stronger correlation values are sometimes found when considering terrain roughness rather than elevation.

Regardless of its simplicity, the classification, when considering the precipitation and temperature grid resolution, is quite
satisfactory for the purposes of this study, especially when compared with the K3 Mountain classification (Karagulle et al., 2017), a much more complex categorization based on many different parameters, obtained from Global Mountain Explorer 2.0 platform (Sayre et al., 2018).

### 2.4   Data analysis techniques

#### 2.4.1   Standardized Precipitation Index (SPI)

Monthly precipitation values are used to calculate the Standardized Precipitation Index (McKee et al., 1993) at 3 and 12 month scale. The index is obtained by fitting the gamma probability distribution $g(x) = x^a e^{-x/b}$ to each month of the year's series of values. To do this, the shape parameter $a$ and the scale parameter $b$ of the gamma distribution are calculated for each series of non-zero values of the same month using the the maximum likelihood method (Choi and Wette, 1969). Afterwards, the cumulative probability $G$ is calculated by applying the formula:

$$G(x_{i,j}) = \int_0^{x_{i,j}} g(x_{i,j})\,dx = \frac{1}{b_j^{a_j}\Gamma(a_j)} \int_0^{x_{i,j}} x_{i,j}^{a_j-1} e^{-x_{i,j}/b_j}\,dx \tag{1}$$


where $j$ is the month index ($j = 1, 2...12$), $i$ is the index for the whole series ($i = 1, 2...n$, with $n$ monthly records), including zero values, and $\Gamma(a_j)$ is the gamma function, i.e. $\int_0^\infty y^{a_j-1} e^{-y}\,dy$.

To take the probability of zero values into account (given that the gamma distribution is defined for $x \in (0, \infty)$), the zero-inflated model is defined as:

$$H(x_{i,j}) = q_j + (1 - q_j)G(x_{i,j}) \tag{2}$$


where $q_j$ is the probability of zero precipitation for the $j$-th month of the year. Finally, the SPI is calculated as the normal inverse function of $H$ via the formula:

$$SPI(x_{i,j}) = -\sqrt{2}\,erfcinv(H(x_{i,j})) \tag{3}$$

where $erfincinv$ is the inverse of the complementary error function:

$$erfc(x) = \frac{2}{\sqrt{\pi}} \int_x^\infty e^{-t^2}\,dt \tag{4}$$


The SPI obtained through this method is thus a series of positive and negative values belonging to a normal distribution ($\mu = 0$, $\sigma = 1$): positive (negative) values represent precipitation above (below) the mean, and an empirical classification of drought





**Table 1.** Classification of SPI/SPEI values and probability of occurrence of each class (World Meteorological Organization, 2012).

| SPI value | Class | Probability |
|---|---|---|
| SPI≥2.00 | Extremely wet | 2.3 |
| 1.50≤SPI<2.00 | Severely wet | 4.4 |
| 1.00≤SPI<1.50 | Moderately wet | 9.2 |
| 0.00≤SPI<1.00 | Mildly wet | 34.1 |
| -1.00<SPI≤0.00 | Mildly dry | 34.1 |
| -1.50<SPI≤-1.00 | Moderately wet | 9.2 |
| -2.00<SPI≤-1.50 | Severely wet | 4.4 |
| SPI≤-2.00 | Extremely wet | 2.3 |

conditions is used (Table 1). To calculate the index at a different time scale, a moving average is first applied to the each monthly value, with length equal to the desired time scale and only previous data included; data that doesn't have enough preceding values to calculate the moving average is discarded. After calculating SPI on this data, each monthly value of the index describes how the conditions for a period with length equal to the time scale and ending in one particular month compares with all others in the series. For example, the SPI at 3 month time scale (SPI-3) for the month of July of a particular year indicates how much dry/wet the the previous 3 months have been compared with all other May-July periods in the series.

The probability distribution chosen for the calculation is the gamma function because, although other possible distributions have been proposed in the literature (Angelidis et al., 2012), including empirical ones (Laimighofer and Laaha, 2022), no single one was shown to be markedly better than the gamma distribution. In-built Matlab® functions, instead of the linear approximations usually presented in the literature (Tigkas et al., 2015; Angelidis et al., 2012; Bordi and Sutera, 2002; Hayes et al., 1999), are used for both the incomplete gamma function and its scale and shape parameters calculation and the calculation of the normal inverse function.

### 2.4.2 Standardized Precipitation Evapotranspiration Index (SPEI)

In order to take into account the effect of evaporative demand on drought episodes, and to compare SPI values, the Standardized Precipitation Evapotranspiration Index (Vicente-Serrano et al., 2010) is calculated, again at 3 and 12 month scale. The procedure for calculating the index is the same as the SPI, but the data analyzed is a series of monthly precipitation minus Reference Evapotranspiration ($ET_0$, in mm) values and a log-logistic probability distribution is used. Temperature data was used to calculate monthly $ET_0$ values using the Hargreaves formula (Hargreaves and Samani, 1985), following the recommendations for SPEI calculation (Beguería et al., 2014). Probability Weighted Moments (PWMs) using Hosking's unbiased method (Hosking, 1986) were used to calculate the $\alpha$, $\beta$ and $\gamma$ parameters of the log-normal distribution for each month of the year, according to the formulae:

$$w_s = \frac{1}{N} \sum_{i=1}^{N} \frac{\binom{N-i}{s} D_i}{\binom{N-1}{s}} \tag{5}$$




$$\beta = \frac{2w_1 - w_o}{6w_1 - w_0 - 6w_2} \tag{6}$$

$$\alpha = \frac{(w_0 - 2w_1)\beta}{\Gamma(1 + 1/\beta)\Gamma(1 - 1/\beta)} \tag{7}$$

$$\gamma = w_o - \alpha\Gamma(1 + 1/\beta)\Gamma(1 - 1/\beta) \tag{8}$$

where $D_i$ are the precipitation minus $ET_0$ values for a given month of the year and $w_s$ are the s order of the PWM. Using the $\alpha$, $\beta$ and $\gamma$ parameters the log-logistic distribution is calculated as:

$$F(D_{i,j}) = \left[1 + \left(\frac{\alpha_j}{D_{i,j} - \gamma_j}\right)^{\beta_j}\right]^{-1} \tag{9}$$

where $j$ indicates the month of the year to which $D_{i,j}$ belongs. Finally, this $F$ distribution is then transformed into a normal
distribution to obtain the SPEI values. Again, moving average windows are applied to the input data in order to obtain different time scales. The distribution was shown to be well suited to analyze the data, as calculations obtained a finite solution for all series.

### 2.4.3   Trend analysis

Annual and seasonal precipitation series, as well as deseasonalized and seasonal maximum and minimum temperature series,
are analyzed in order to search for significant (at 5% significance) trends. Drought index series are also analyzed in order to search for trends, but the autocorrelation of the series (given that a moving average is applied to the monthly series in order to obtain the 3 and 12 month scale data) is taken into account by also applying different pre-whitening methods. These method are the simple Pre-Whitening method (PW, Kulkarni and Storch, 1995), the Trend Free Pre-Whitening method (TFPW, Yue et al., 2002) and the Variance Corrected Trend Free Pre-Whitening method (VCTFPW, Wang et al., 2015). The different results
are used to obtain one trend evaluation by applying the 3PW algorithm (Collaud Coen et al., 2020), which, for the purpose of this study, is as follows:

1. If the lag-1 autocorrelation of the data is significant, the PW and TFPW series are obtained from the original series.

2. The trend is considered significant if both processed series return significant trends; the significance is chosen as the lower of the PW and TFPW series calculated via the Mann-Kendall test.

3. The slope of the significant trend is given as the Sen's slope of the VCTFPW series.

If the lag-1 autocorrelation of a series is found not to be significant, trend analysis is performed on the un-modified data.



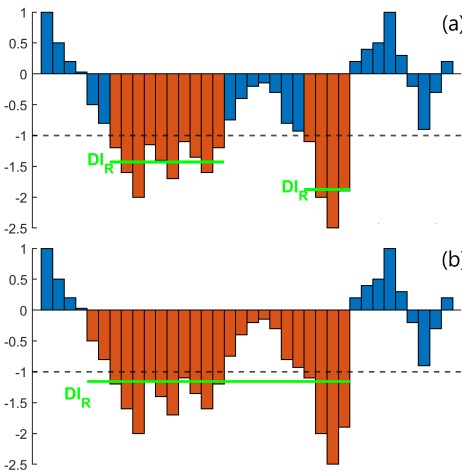

**Figure 3.** Drought runs examples, highlighted in orange. **(a)** Runs obtained with a simple -1 threshold. **(b)** Run defined by a threshold and by including including their onset and offset, meaning the negative values before and after the values under the threshold. In both cases $DS_R$ is the sum of the index value during the run, $DD_R$ is the length of each run and $DI_R$ is the mean index value during the run.

### 2.4.4 Run analysis

Each SPI/SPEI series, at both time scales, is analyzed through *run analysis*. Based on run theory (Yevjevich, 1967), run analysis defines a drought run as a series of consecutive months under a certain threshold (-1, corresponding to a moderately dry condition in the SPI classification). Adding to this definition, the negative values leading and following a period under the -1 threshold are counted as part of the runs, in order to capture events where a deficit is not fully recovered from. The differences between the use of a single threshold and the inclusion of the run onset and offset are shown in Figure 3. Drought run characteristics are then calculated for each run (Caloiero et al., 2021):

- Drought Duration ($DD_R$) is the length of the drought run, reported in months for this study;

- Drought Severity ($DS_R$) is the cumulative value of the index during each run;

- Drough Intensity($DI_R$) the ratio between the $DS_R$ and $DD_R$ value of a run, i.e. the average index value during the run.

Given a series of drought runs calculated from an index series, the average value of these characteristics for all runs are reported as $\overline{DD_R}$, $\overline{DS_R}$ and $\overline{DI_R}$.

### 2.4.5 Event analysis

In order to study the spatio-temporal characteristics of drought, and to define region-wide drought events, drought indices are analyzed in a similar way to that proposed in González-Hidalgo et al. (2018), Baronetti et al. (2020) and Baronetti et al. (2022).





Drought episodes are detected through the use of two thresholds (the minimum duration criteria of 3 weeks, used in the cited papers, is discarded as monthly data is used in this analysis):

- an index threshold, based on the classification of SPI/SPEI (see Table 1): cells with and index lower than -1 are considered to be in drought condition;

- an area threshold: a drought episode is considered in progress when at least 25% of the domain is in drought condition.

In addition to these two thresholds, the drought episodes' *onset* and *offset*, meaning the periods below the 25% drought area threshold before and after a period above the threshold, were also included in the the drought period itself. This approach is useful in considering persisting drought conditions as one continuous event while still maintaining well-defined episodes, similar to the proposed drought run definition (see Figure 3).

To contrast with the drought runs, which are calculated from a series of index values belonging to one cell, episodes evaluated through this analysis are called *drought events*. So, for example, during a drought event a certain percentage of the domain will be in drought conditions (below the -1 drought index threshold); each of these cells will therefore be experiencing a drought run. Similarly to drought runs, different characteristics are calculated for each drought event:

- Drought event Duration ($DD_E$) is the length of the drought event;

- Drought event Severity ($DS_E$) is the sum of the drought index of each cell in drought condition for the duration of the event, divided by the total number of cells in the domain;

- Drought event Intensity ($DI_E$) is the mean of the local intensity for each cell that has been part of the drought event. Intensity for each cell is calculated as the sum of the drought index below the -1 threshold divided by the number of months where the index was lower than -1.

- Drought Area ($DA_E$) is the average number of cells in drought condition during the event;

Given a series of drought events calculated from the index series for all cells in the domain, the average value of these characteristics for all events are reported as $\overline{DD}_E$, $\overline{DS}_E$, $\overline{DI}_E$ and $\overline{DA}_E$.

### 2.4.6 Two sample t-test

In order to evaluate the significance of the change in average drought characteristics between the periods 1958-1990 and 1990-2023 (approximately the first and second half of the series), the two sample t-test (Rasch et al., 2011) is applied to $\overline{DS}_R$, $\overline{DD}_R$ and $\overline{DI}_R$ (as well as $\overline{DS}_E$, $\overline{DD}_E$, $\overline{DI}_E$ and $\overline{DA}_E$) calculated for the runs starting before and after January 1990, respectively. After obtaining the values pre and post 1990, their sample mean and standard deviations are calculated, and the test statistic $t$ is calculated as

$$t = \frac{\overline{DC}_{post} - \overline{DC}_{pre}}{s} \tag{10}$$





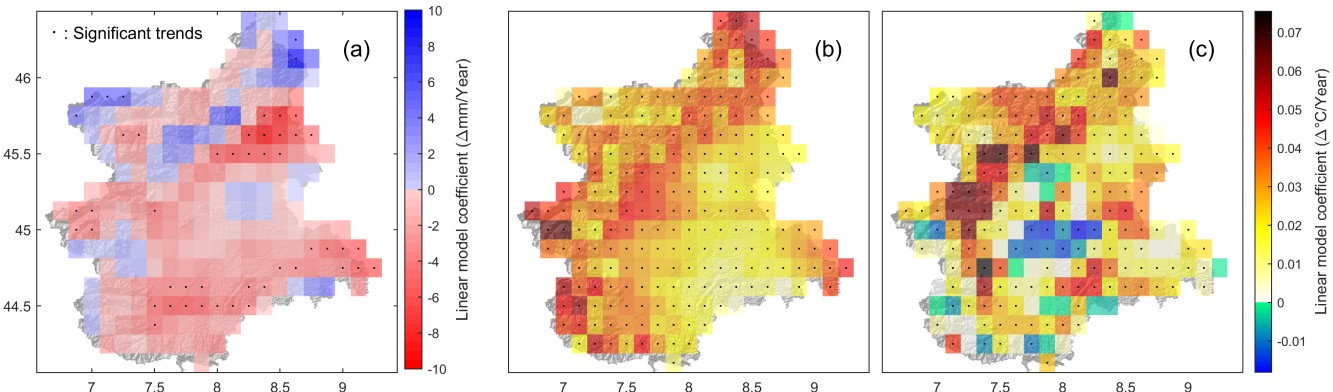

**Figure 4.** Results of trend analysis on meteorological data for the Piedmont and Aosta valley region. **(a)** Annual precipitation trends, **(b)** maximum monthly temperature trends, **(c)** minimum monthly temperature trends. Cells containing a dot denote significant trends at 5% significance.

where $\overline{DC}$ is the mean of a certain drought characteristics for all drought runs/events before/after 1990 and $s$:

$$s = \sqrt{\frac{\sigma_{post}^2}{n_{post}} + \frac{\sigma_{pre}^2}{n_{pre}}} \tag{11}$$

where $\sigma$ is the standard deviation of a certain drought characteristics for all drought run/events before and after 1990 and $n$ the number of run/events for the two periods. $t$ is then compared with the critical value of the statistic at a 5% significance

level. Given that no assumptions about the variance of the two ensembles were made, and given the different number of runs in the two periods, the degrees of freedom needed for the calculation of the critical value were approximated through the Welch-Satterthwaite equation (Welch, 1947).

## 3 Results

### 3.1 Precipitation and temperature trends

The precipitation and temperature data used to calculate the SPI/SPEI is first analyzed in order to find possible trends, so as to contextualize the results obtained from the drought indices. For the most part, changes in both annual and seasonal precipitation values for the Piedmont and Aosta Valley region are found not to be statistically significant (Figure 4). Only some portions of the region, bordering on the Alpine chain in the north and south, show significant trends, mostly reporting reduced precipitation. The only difference is found in the winter season (shown in supplementary material B1), where 30% of cells, covering a large

part of the south half of the region, report decreases as high as -2.41 mm/year, but mostly around -1.2 mm/year. Still, mean values of precipitation for the whole region do not show a statistically significant decrease either at the annual or seasonal scale.



In contrast to precipitation values, all of the region is found to be affected by significant temperature trends. Maximum temperature is rising all over the domain, from 0.008 to 0.065 °C/year, with a mean of 0.03 °C/year; this rate of increase is significantly correlated with altitude (0.38 correlation value), although a small zone, in the flat part of the region on the western border, shows slope coefficients comparable with those at higher altitudes.

Minimum temperatures, while overall still increasing, present less homogeneous results. First, 6.6% of cells, mainly grouped at the center of the region, show a negative trend; furthermore 16% of cells show no significant trend. Minimum temperature change ranges from -0.02 to 0.07 °C/year, with a mean of 0.02 °C/year, and also shows a significant correlation with altitude, although with a lower 0.16 correlation value. Seasonal analysis largely reflects the results for both minimum and maximum annual temperature, while showing the highest seasonal increases in winter (shown in supplementary Figure B2 and B3).

Coherent with local results (Figure 4), maximum and minimum temperatures for the whole region are found to be increasing at a rate of 0.03 and 0.02 °C/year, respectively.

The reported results, both for the precipitation and temperature trends, are in agreement with an analysis made on the same data set by the local environmental agency ( *Arpa Piemonte*, 2020a). More importantly, the analysis is coherent with the results obtained from studies conducted on the Piedmont and Aosta Valley meteorological station network data (Ciccarelli et al., 2008; Acquaotta et al., 2009), and particularly on high elevation stations' data (Acquaotta et al., 2015; Terzago et al., 2013).

## 3.2 Drought indices trends

Drought indices calculated from the precipitation and temperature series of each cell in the region (see Figure 5) are analyzed in order to find possible trends in drought conditions. Given the nature of SPI, as described in Section 2.4.1, negative trends indicate a tendency for precipitation to be below the series's mean value. This means that both wet and dry periods have seen on average reduced precipitation, and thus that droughts conditions, when occurring, have become worse. For the SPEI, described in Section 2.4.2, the trend interpretation is the same, but instead of precipitation a climatic water balance between precipitation and potential evapotranspiration is considered. Furthermore, trend analysis on indices at the shorter 3 month time scale and the longer 12 months time scale indicates, respectively, how drought conditions might have evolved over smaller time scales, generally correlated with soil moisture, and over larger time scales, generally correlated with water reservoirs and groundwater levels.

Trend analysis on SPI-3 and SPI-12 values shows results that mostly agree with the trends in annual precipitation, with cells with significant decreases in index values (and thus a tendency towards dryer conditions) almost completely overlapping with those with significant annual precipitation trends. A majority of cells (almost 70%) shows trends at both 3 and 12 months time scale, although SPI-12 indicates overall worse conditions over time compared to SPI-3.

Trend analysis for SPEI-3 and SPEI-12 displays a similar time scale effect, with the longer time scale having a higher number of cells with significant trends (although with 79% of cells showing trends for both time scales) and, on average, greater slope coefficients.





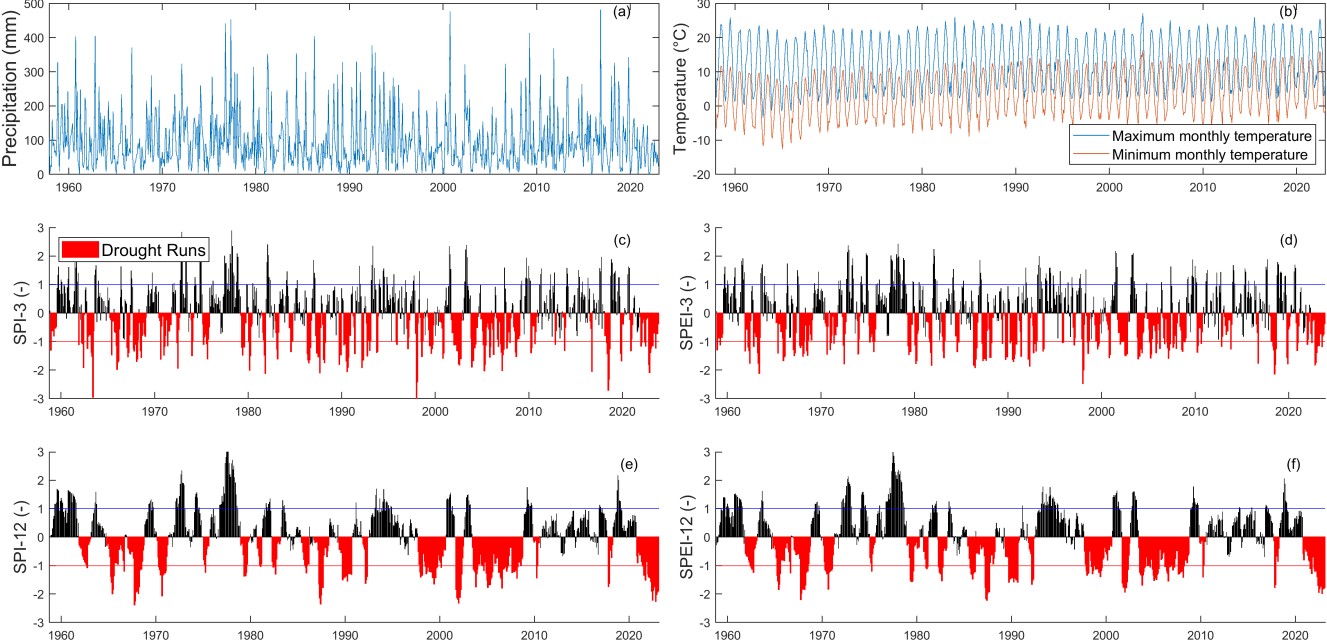

**Figure 5.** Meteorological data and drought indices series for a representative point in the domain. **(a)** Precipitation, **(b)** maximum and minimum temperature, **(c)** SPI-3, **(d)** SPEI-3, **(e)** SPI-12, **(f)** SPEI-12.

The clearest differences emerge by comparing the results obtained by SPI and SPEI. The latter shows, at both time scales, a far wider region heading towards dryer conditions (more than twice the cells found with SPI), and at a faster rate, given the slope coefficients. Furthermore, a clear relation between orography and trends becomes apparent from the SPEI analysis, seen as, contrary to the mountainous areas, most of the flat part of the region is affected by significant trends (Figure 6).

The importance of terrain characteristics in determining the significance of trends is also evaluated by performing a trend analysis on the data belonging to four different areas, namely the flat, hilly, and mountainous areas (as defined in Section 2.3). The drought indices, calculated from cumulative precipitation and mean temperature for the four areas, show a clear difference in observed trends. The flat part of the region reports significant drying trends for both SPI-12, SPEI-3 and SPEI-12, despite the trend in annual precipitation being not significant and the temperature trends having a lower slope coefficient than at higher altitudes (Figure 7). Conversely, the alpine chain reports no significant trends in the indices, even though the temperature reports higher slope coefficients than the other areas. As a comparison, the same trend analysis is performed on data belonging to areas defined by mean elevation thresholds (also defined in Section 2.3) finding largely the same results: the two lower elevation areas still report significant drying trends for SPEI-12 (but not for SPI-12 and for SPEI-3, the latter being significant only in the second lowest elevation area) and no significant precipitation trend, while temperature trends remain significant and with higher coefficients at higher elevation areas. Overall, taking the results obtained both with ruggedness-defined areas and




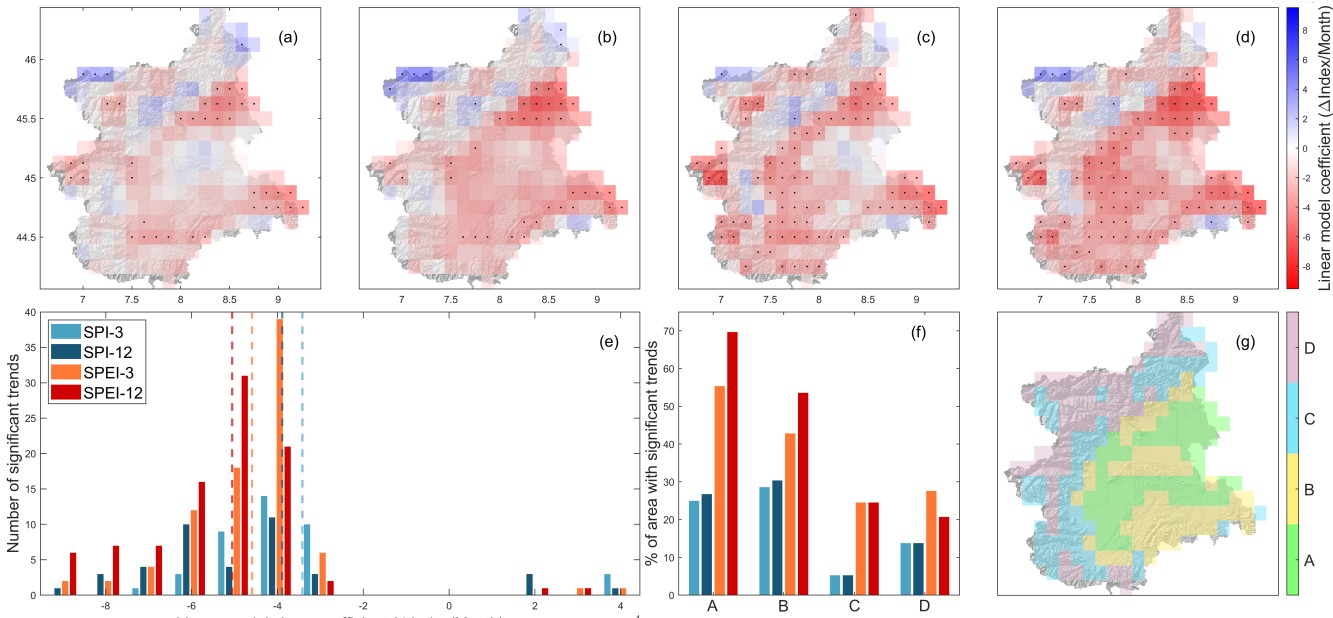

**Figure 6.** Trend analysis on drought indices. **(a)** SPI-3 trends. **(b)** SPEI-3 trends. **(c)** SPI-12 trends. **(d)** SPEI-12 trends. Cells containing a dot denote significant trends at 5% significance. **(e)** Distribution of trends' Sen-slope coefficients with dashed lines representing the respective mean values. **(f)** Distribution of terrain roughness for cells with significant trends. **(g)** Representation of terrain roughness classes (see Figure 2 for more details on their definition).

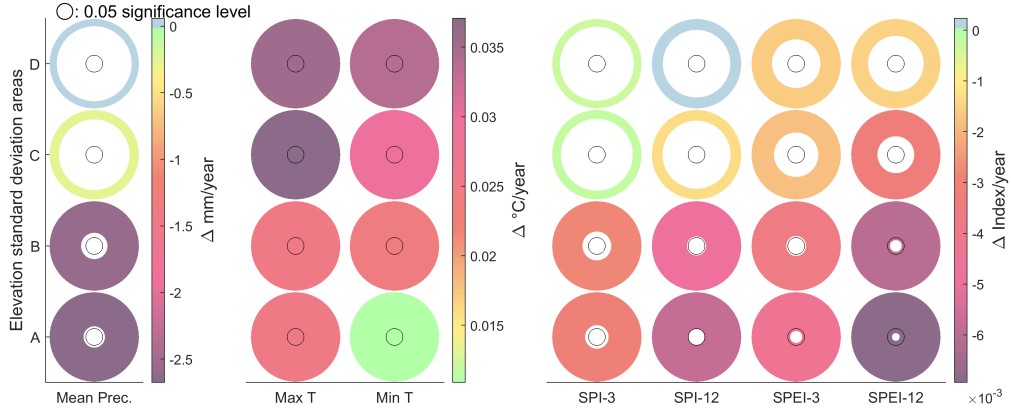

**Figure 7.** Trend analysis on mean annual precipitation, temperature and drought indices calculated from data belonging to areas defined by terrain roughness inside cells. The colour of the circles represents the slope coefficient of the trend, while the inner radius of the circles represents the significance of the trend (a smaller inner radius represents a more significant trend). The black circles denote a significance level of 5%. A, B, C and D areas are displayed in Figure 6



with elevation-defined areas, a significant difference between the alpine range and the plain area of the domain is observed, the trend analysis reporting worsening drought conditions for the latter.

## 3.3   Drought run analysis

After analyzing how the general drought conditions in the region have changed over time, the effects of such changes on the characteristics of local drought periods are investigated through *run analysis* (see Section 2.4.4). First, the characteristics of
drought runs in the region are described as a baseline; then, possible changes in the number, severity, duration and intensity of the runs are investigated.

### 3.3.1   Drought runs characteristics

Despite the differences in observed trends detailed in the previous section, the detection of drought runs by SPI and SPEI shows a high level of correspondence for both 3 and 12 months time scales. After changing each cell's series into a binary
series of zeros and ones, were 1 denotes the occurrence of a drought run, Cohen's kappa (Cohen, 1960) between the series is calculated as a measure of agreement. The mean Kappa value is slightly higher at 3 month scale (mean kappa equal to 0.86) than at 12 month scale (mean kappa equal to 0.81), but always higher than 0.5: this means that there is always good to excellent agreement between the identification of drought runs based on SPI and on SPEI. Therefore, given that SPI runs are based only on precipitation values, it can be stated that a majority of drought runs are determined by precipitation deficits, with
temperature itself having a smaller influence on single events, and a greater influence on overall trends.

In the analysis of drought run characteristics, the longer time scale shows, both for SPI and SPEI, a lower number of runs, but with higher severity ($DS_R$) and duration ($DD_R$). This difference is expected, as can be seen from a visual inspection of the drought runs at different time scales (see Figure 5). Drought intensity ($DI_R$) values are instead similar between the two time scales, although slightly greater at the 3 month scale. Despite the difference in absolute values, drought analysis indicates a
higher number of runs, a higher $\overline{DS_R}$ and $\overline{DD_R}$ for SPEI runs compared to SPI runs at both time scales. Thus, when considering both precipitation and $ET_0$, a greater number of longer and more severe drought events are detected, compared to the less numerous and shorter precipitation-only events.

One significant difference emerges when comparing the $\overline{DI_R}$ values. Average drought intensity is lower for SPEI-3 runs compared to SPI-3, while slightly greater for SPEI-12 compared to SPI-12. This seems to be due to the lower index values
reported by SPI-3 compared with SPI-12. The mean negative minimum values during the drought runs are lower for SPI than for SPEI at both 3 months ($-1.65 \pm 0.05$ for SPI and $-1.47 \pm 0.03$ for SPEI) and 12 months ($-1.52 \pm 0.07$ for SPI and $-1.46 \pm 0.06$ for SPEI) time scales, but, while SPEI values remain almost constant, SPI values show less negative mean minimum values at the longer time scale. This fact, combined with the similarly longer SPEI runs at both time scales leads to the slightly higher $\overline{DI_R}$ for SPEI-12.

Regarding the spatial distribution of mean drought run characteristics (number of runs, $\overline{DS_R}$, $\overline{DD_R}$ and $\overline{DI_R}$), SPI and SPEI show similar results when compared at the same time scale, while deviating significantly between 3 and 12 months time





**Table 2.** Correlation coefficients and relative p-values between mean drought characteristics and elevation, for both SPI and SPEI at 3 an 12 month time scale.

| | Number of runs | | $\overline{\text{DS}}_\text{R}$ | | $\overline{\text{DD}}_\text{R}$ (months) | | $\overline{\text{DI}}_\text{R}$ | |
|---|---|---|---|---|---|---|---|---|
| | C | p-value | C | p-value | C | p-value | C | p-value |
| **SPI-3** | -0.50 | $1.78 \times 10^{-15}$ | -0.35 | $6.60 \times 10^{-8}$ | 0.44 | $3.03 \times 10^{-12}$ | 0.20 | $2.54 \times 10^{-3}$ |
| **SPEI-3** | -0.29 | $7.84 \times 10^{-6}$ | -0.15 | $2.22 \times 10^{-2}$ | 0.14 | $3.35 \times 10^{-2}$ | 0.17 | $8.81 \times 10^{-3}$ |
| **SPI-12** | 0.06 | $3.65 \times 10^{-1}$ | 0.13 | $5.62 \times 10^{-2}$ | -0.11 | $9.24 \times 10^{-2}$ | 0.35 | $3.86 \times 10^{-8}$ |
| **SPEI-12** | 0.04 | $5.41 \times 10^{-1}$ | 0.05 | $4.23 \times 10^{-1}$ | -0.07 | $3.14 \times 10^{-1}$ | 0.18 | $6.84 \times 10^{-3}$ |

scales. SPI-3 and SPEI-3 run characteristics do not display any spatial gradient, but do display some correlation with elevation (Table 2). In particular, when areas at higher elevation are considered, more severe, longer and less intense runs are reported (although with some differences in the degree of correlation). All together, these results indicate that, on shorter time scales, drought runs in higher elevation areas tend to be more clustered. Even so, visual inspection of the spatial distribution of drought characteristics for SPI-3 and SPEI-3 (see supplementary Figure C1) indicates that, while the higher elevation points of the mountainous part of the domain do show quite uniform drought characteristics consistent with the observed correlations, spatial variability of characteristics is overall quite high. It can therefore be stated that, despite some significant effects of elevation on the characteristics of drought periods, local orography and meteorological conditions play a key role.

SPI-12 and SPEI-12 run characteristics display no spatial gradient (see supplementary Figure C2) and no correlation with altitude in terms of number, severity and duration of runs. The only statistically significant correlation found is with $\overline{\text{DI}}_\text{R}$, with higher elevation areas reporting less intense events, coherent with the results obtained for SPI-3 and SPEI-3.

Possible correlations of drought characteristics with terrain ruggedness are also considered (Table 3), but the resulting correlation values are always similar or lower than those found with elevation for the 3 month scale. Conversely, indices at the 12 month scale reported significant correlations for the number of runs and their $\overline{\text{DS}}_\text{R}$ and $\overline{\text{DD}}_\text{R}$, with rugged terrain reporting less numerous, less severe and shorter drought runs.

### 3.3.2 Temporal analysis of drought run characteristics

Trend analysis on the obtained drought runs reports only a few cells (always less than 3% of the domain) showing significant trends for $\text{DD}_\text{R}$, $\text{DS}_\text{R}$ and $\text{DI}_\text{R}$. In comparison, SPEI-3 shows a far greater amount of cells, slightly more than 10% of the total area, with significant increasing trends for $\text{DS}_\text{R}$ and $\text{DI}_\text{R}$, distributed almost exclusively along the alpine chain, particularly near the southern border. The yearly predicted change, in terms of percentage of the relative $\overline{\text{DS}}_\text{R}/\overline{\text{DI}}_\text{R}$ for the cell, ranges from 1 to 11% and 0.01 to 1% for severity and intensity respectively.

Despite the overall lack of significant trends, clear differences can be found between the characteristics of drought runs that started before and after 1990, approximately at half the series' length. SPI-3 and SPEI-3 display on average an increase in





**Figure 8.** Cells with significant changes in mean drought characteristics between the 1958-1990 and 1990-2023 periods according to the two sample t-test. **(a)** $\overline{DS}_R$ increase, **(c)** $\overline{DS}_R$ decrease, **(b)** $\overline{DD}_R$ increase, **(d)** $\overline{DD}_R$ decrease.





**Table 3.** Correlation coefficients and relative p-values between mean drought characteristics and terrain roughness, for both SPI and SPEI at 3 an 12 month time scale.

|  | Number of runs | | $\overline{\mathrm{DS}}_\mathrm{R}$ | | $\overline{\mathrm{DD}}_\mathrm{R}$ (months) | | $\overline{\mathrm{DI}}_\mathrm{R}$ | |
|---|---|---|---|---|---|---|---|---|
|  | C | p-value | C | p-value | C | p-value | C | p-value |
| **SPI-3** | -0.40 | $6.74\times10^{-10}$ | -0.22 | $1.06\times10^{-3}$ | 0.29 | $7.18\times10^{-6}$ | 0.26 | $9.25\times10^{-5}$ |
| **SPEI-3** | 0.22 | $9.87\times10^{-4}$ | -0.06 | $3.64\times10^{-1}$ | 0.02 | $7.92\times10^{-1}$ | 0.19 | $3.86\times10^{-3}$ |
| **SPI-12** | 0.18 | $6.77\times10^{-3}$ | 0.23 | $3.68\times10^{-4}$ | -0.21 | $1.52\times10^{-3}$ | 0.31 | $2.44\times10^{-6}$ |
| **SPEI-12** | 0.19 | $4.73\times10^{-3}$ | 0.19 | $3.24\times10^{-3}$ | -0.21 | $1.61\times10^{-3}$ | 0.12 | $7.25\times10^{-2}$ |

the number of runs (more markedly in the case of SPEI-3), and in their $\mathrm{DI}_\mathrm{R}$. Opposite results are found in terms of $\mathrm{DS}_\mathrm{R}$ and $\mathrm{DD}_\mathrm{R}$, with SPI-3 indicating a shift towards less severe and shorter drought runs, and vice-versa for SPEI-3. Significantly, this difference seems to be caused mainly by cells located in the flat part of the region, where SPEI-3 indicates a shift towards greater $\mathrm{DS}_\mathrm{R}$ and $\mathrm{DD}_\mathrm{R}$. The rest of the region shows similar results for the two indices. The alpine chain, especially in the north, shows a shift towards a higher number of less severe, shorter and less intense drought runs. SPI-12 and SPEI-12, on the other

hand, report agreeing results and show on average a change towards a lower number of more severe, longer and more intense runs across the domain. The only exception is the alpine chain, where for a small but continuous area a change towards less numerous, less severe, shorter and less intense runs is found.

These relative changes are highly correlated with the ruggedness of the area (Table 4). For example, at the 3-month scale, the flat part of the region has seen a change towards less numerous, more severe, longer and more intense drought runs, while

the alpine chain shows an opposite change. Changes in SPEI-12 run characteristics also display a similar correlation for $\overline{\mathrm{DS}}_\mathrm{R}$, $\overline{\mathrm{DD}}_\mathrm{R}$ and $\overline{\mathrm{DI}}_\mathrm{R}$ but opposite in terms of number of runs. Therefore, it seems that SPEI-12 runs got more numerous, more severe, longer and more intense in the lowlands, and, although not quite as stark, the opposite has happened in the alpine chain. SPI-12 does show an increase in the number, severity and duration of drought runs in the lowlands and a decrease in the mountains, but no correlation for $\overline{\mathrm{DI}}_\mathrm{R}$.

Opposite to average drought run characteristics, changes in drought run characteristics report higher correlation with terrain ruggedness than with elevation. Overall, correlation values are also higher than those found for average drought run characteristics, and visual inspection of the spatial distribution does show a quite homogeneous distribution of drought run characteristics change between the mountains (especially on the windward side, i.e. the one facing the Po plain) and the plains and hills. The only outliers are the Aosta valley and another valley close to the western border, with changes often in common with the

lowlands.

Still, most of the changes found by comparing the two periods are not found to be significant according to the two sample t-test, and thus do not denote a change in the statistical distribution of drought run characteristics. The cells with significant





**Table 4.** Correlation coefficients and relative p-values between change in mean drought characteristics pre- and post-1990 and terrain roughness, for both SPI and SPEI at 3 an 12 month time scale.

| | $\Delta$**Number of runs** | | $\Delta\overline{\mathbf{DS}}_\mathbf{R}$ | | $\Delta\overline{\mathbf{DD}}_\mathbf{R}$ | | $\Delta\overline{\mathbf{DI}}_\mathbf{R}$ | |
|---|---|---|---|---|---|---|---|---|
| | C | p-value | C | p-value | C | p-value | C | p-value |
| **SPI-3** | 0.18 | $5.22\times10^{-3}$ | 0.42 | $3.07\times10^{-11}$ | -0.37 | $8.66\times10^{-9}$ | 0.41 | $1.90\times10^{-10}$ |
| **SPEI-3** | 0.23 | $5.57\times10^{-4}$ | 0.38 | $2.30\times10^{-9}$ | -0.38 | $4.11\times10^{-9}$ | 0.26 | $6.39\times10^{-5}$ |
| **SPI-12** | -0.33 | $3.18\times10^{-7}$ | 0.25 | $1.79\times10^{-4}$ | -0.27 | $4.94\times10^{-5}$ | 0.09 | $1.67\times10^{-1}$ |
| **SPEI-12** | -0.25 | $1.70\times10^{-4}$ | 0.32 | $1.05\times10^{-6}$ | -0.30 | $3.95\times10^{-6}$ | 0.28 | $1.85\times10^{-5}$ |

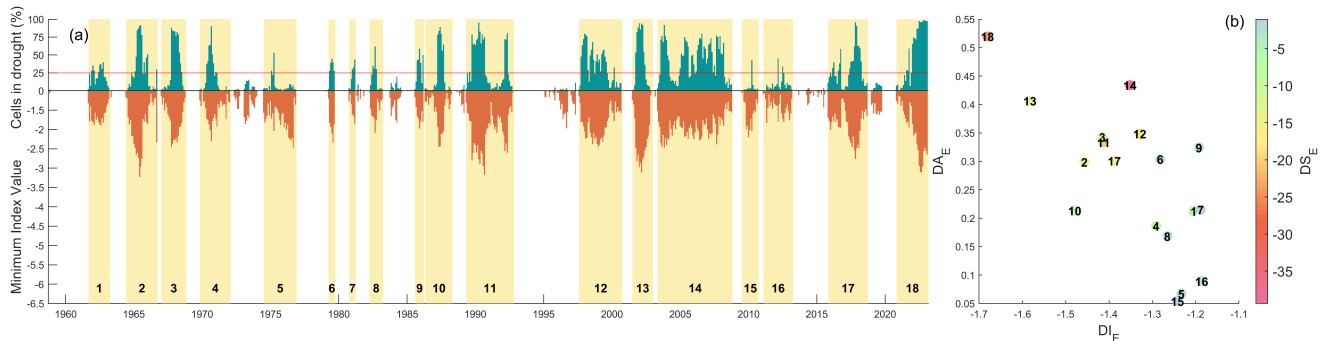

**Figure 9.** Drought event analysis conducted on SPEI-12. **(a)** Series of percentage of cells in drought condition (below the -1 threshold) and the minimum index value in the domain. Each event is highlighted in yellow and labeled. **(b)** Drought event characteristics.

changes (reported in Figure 8) are mostly distributed between two areas. Changes towards more severe (according to SPI-12, SPEI-3 and SPEI-12), longer (by both indices at the 3 month scale) and more intense (by both indices at the 12 month scale) drought runs are reported for the eastern-most part of the domain. Changes towards less severe and shorter drought runs are reported mostly in the northern part of the alpine chain for SPI/SPEI at the 3 month scale, while almost no significant shifts towards less intense runs are reported.

## 3.4 Drought event analysis

This section introduces the results obtained from the analysis of drought events, i.e. droughts periods on a region-wide scale (see Section 2.4.5). Similarly to the previous section, both the characteristics of drought events and their change over time are discussed.



### 3.4.1 Drought event characteristics

Drought events are calculated from SPI and SPEI index series at 3 and 12 month scale. The analysis displays similar results between the two indices at the same time scale, with all main events identified by both SPI and SPEI, and high agreement
between the extent of the area in drought conditions over time.

The analysis at the 3 month scale reports about 60 events (see supplementary Figure D1), while the analysis at the 12 month (for SPEI-12 see Figure 9, for SPI-12 see supplementary Figure D2) scale reports less than 20. Drought events at the longer time scale are also far more severe and longer than those at the shorter time scale, but intensity and area values are similar. Regarding relative differences between the drought characteristics between SPI and SPEI at both time scale, $\overline{DS}_E$ is
similar between the two indices , $\overline{DD}_E$ is higher for SPEI, and both $\overline{DI}_E$ and $\overline{DA}_E$ are higher for SPI. On the other hand, when considering the mean highest area affected by drought conditions in every single event both indices report similar results at both time scales. Overall, this indicates that the same deficit tends to affect a slightly wider area, with a higher intensity but for less time when only precipitation is considered, while it tends to affect the same overall area with less intensity and for a longer time when both precipitation and temperature are considered.

Drought event analysis on SPI-12 and SPEI-12 was also useful in indicating the main drought events that happened in the region in the last 60 years. Of these, the last one, starting in the winter of 2021 and still ongoing, was identified as perhaps the most extreme in the series. In particular, the wide area affected by drought during this event and its severity, second only to the longest 2001-2008 event, mark it as an exceptional drought for the region. The intensity value is also the highest of all detected events, but this may not be significant given that this last event has not yet ended. Certainly, the fact that its severity is higher
than the severity of the 2001-2002 event as detected through SPEI-12, also adds to how exceptional this last event is.

### 3.4.2 Temporal analysis of drought event characteristics

Trend analysis reports no significant results for the drought characteristics obtained through event analysis. Confronting the values before and after 1990 does show results consistent with those found for the drought runs (Table 5): drought events have become more severe, longer and more intense at both time scales. Also similar to drought runs, the number of drought events
has increased at the shorter 3 month time scale while it has decreased at the longer 12 month scale. Another difference is in the $\overline{DA}_E$, which has decreased at the 3 month time scale and has increased at the 12 month time scale. Overall, this seems to indicate that, on a region wide level, drought conditions have worsened between the periods 1960-1990 and 1990-2000, with short term deficits becoming more common over slightly smaller areas, leading to more generalised deficits over wider areas at the longer time scales. Despite many of the described changes not being significant according to the two sample t-test, $\overline{DS}_E$
and $\overline{DD}_E$ for SPI-12 do report a statistically significant shift in the mean before and after 1990. Changes in $\overline{DS}_E$ and $\overline{DD}_E$ for SPEI-12 also report significance levels close to the 5% level, although not achieving the 5% threshold. This seems to confirm that the shift towards worse drought event conditions has worse effects at longer time scales, and that this shift is mainly caused by a change in precipitation patterns.





**Table 5.** Drought event characteristics before and after 1990. Values in bold font denote significant differences between the two distributions at 5% significance.

| | Number of events | | $\overline{\text{DS}}_{RE}$ | | $\overline{\text{DS}}_{E}$ (months) | | $\overline{\text{DS}}_{E}$ | | $\overline{\text{DS}}_{E}$ (%) | |
| --- | --- | --- | --- | --- | --- | --- | --- | --- | --- | --- |
| | Pre 1990 | Post 1990 | Pre 1990 | Post 1990 | Pre 1990 | Post 1990 | Pre 1990 | Post 1990 | Pre 1990 | Post 1990 |
| **SPI-3** | 31 | 28 | -3.24 | -3.30 | 5.68 | 5.90 | -1.45 | -1.50 | 38.51 | 35.13 |
| **SPEI-3** | 27 | 32 | -3.10 | -3.40 | 6.18 | 6.56 | -1.32 | -1.36 | 37.50 | 32.05 |
| **SPI-12** | 12 | 5 | **-7.14** | **-19.34** | **14.33** | **36.60** | -1.37 | -1.42 | 31.61 | 32.05 |
| **SPEI-12** | 11 | 7 | -6.92 | -16.14 | 20.27 | 32.28 | -1.31 | -1.39 | 24.24 | 30.77 |

Despite the apparent importance of precipitation, the only significant trend in terms of the percentage of the domain in
drought conditions (index lower than -1) over time is found for SPEI-12, with a slope coefficient of $2.92 \times 10^{-4}$/year .

## 4   Discussion and conclusion

In this study, 60 years of precipitation and temperature data are analyzed in order to characterise changes in drought conditions in the Piedmont and Aosta valley area, reaching the following conclusions:

1. Trend analysis on temperature and precipitation indicates the presence of a temperature increase in the region—despite
sometimes markedly different results between minimum and maximum temperatures—while few precipitation trends
   are found. Precipitation changes are largely related to season, with clear differences between the winter months and the
   rest of the year, and seem to affect only small portions of the territory, not leading to a region-wide significant trend.

2. Evidence of widespread drying trends in the region is found through the trend analysis of SPI and SPEI series. Tempera-
   ture plays a key role in defining these drying trends, as the SPEI reports negative trends for wider areas and with greater
slope coefficients than SPI. Still, the areas showing the more severe drying trends do not coincide with the areas showing
   the highest warming rates, indicating that changes in droughts are governed by the interplay between temperature and
   precipitation.

3. The start and end of single drought periods seem to be mainly determined by precipitation anomalies, in contrast to
   the importance of temperature in determining long-term conditions. Both local and region-wide drought periods are
identified by both SPI and SPEI, meaning that, at least, temperature alone does not seem to be able to determine drought
   conditions in the absence of precipitation deficits.

4. Some evidence of an increase in the severity, duration and intensity of drought periods after 1990 is found, although
   often not statistically significant. A tendency for drought periods at 3 month time scale to become more numerous, and





for drought periods at 12 month time scale to become less numerous is observed, both at a local and regional scale. Thus, while the percentage of time under drought conditions has become greater at both time scales, it seems that a larger amount of short-term deficits aggregate into long-term deficits with higher duration. In addition to this, a significant positive trend in the percentage of the area under drought conditions according to SPEI-12 is detected.

5. Changes in the characteristics of local drought periods are affected by temperature increase, as drought periods obtained from SPEI series show more pronounced increases in severity, duration and intensity than those obtained from SPI series. Contrary to this, drought events at a region-wide scale show more marked shifts in severity and duration for SPI than for SPEI, denoting a more significant influence of regional precipitation patterns than of temperature on droughts at a regional scale.

6. Terrain characteristics and elevation show significant influence on the observed trends and changes in drought characteristics, with drying trends being more severe the lower and less rugged the area. In fact, when the mountainous parts and the flat part of the domain are considered separately, the first shows no significant drought trends, while the second reports significant drying trends for both SPI and SPEI at multiple timescales. In the case of drought period characteristics, decreases in their severity, duration and intensity are mostly found in the alpine range, while increases are mostly found in the smoother and low lying areas.

Overall, whereas the worsening of drought conditions related to precipitation and temperature for the region is clear, how these changes may affect the characteristics of single drought periods remains less clear. Despite changes between the two halves of the analysed series being statistically significant, almost no trends could be detected. As such, it is difficult to asses whether the increase in severity, duration and intensity of drought periods, although coherent with the worsening of drought conditions, is actually part of a general tendency, and is not just determined by the extreme drought events present in the second half of the series.

Finally, although strong correlations between drought trends and the elevation and ruggedness of the terrain are found, attribution of these results to physical phenomena is not straightforward. Given the importance of precipitation in determining the onset of drought periods, the increase in alpine summer precipitation due to the increase of convective rainfall (Giorgi et al., 2016; Grose et al., 2019) could determine milder drought periods and even the absence of drying trends/presence of wetting trends. On the other hand, widespread downward trends in winter precipitation, as well as somewhat enhanced winter warming trends in the same areas make this explanation not completely satisfactory. Therefore, further research is needed to study how the drought conditions of areas at different elevations and with different reliefs may evolve under climate change.

*Data availability.* The data that support the findings of this study are openly available in the NWIOI data set at https://www.arpa.piemonte.it/rischinaturali/tematismi/clima/confronti-storici/dati/dati.html, maintained and updated by the Forecast Systems Department of the Regional Environmental Protection Agency of Piedmont (*Dipartimento Sistemi Previsionali - Arpa Piemonte*).





## Appendix A: Optimal Interpolation Method

As discussed in Section 2.2, the Optimal Interpolation Method was used by *Arpa Piemonte* to develop the NWIOI dataset used in the present work. The analysis method is described in detail in Uboldi et al. (2008), but information regarding the specific choices made for the dataset are reported in the following paragraphs.

The method produces a regular grid of data by interpolating the station data at each time step. In particular, the interpolation is done by modifying a previously obtained *background* or *first-guess* field available at the desired grid points via a linear relation with the *innovation vector*, meaning the difference between the observed values and the background field values at station points:

$$x_a = x_b + K(y_0 - y_b) \tag{A1}$$

where the vector $x_a$ contains the interpolated (or analyzed) values at each grid node, the vector $x_b$ contains the background field values at each grid point, the vector $y_0$ contains the observed values at each station point and the vector $y_b$ contains the background field values at each station point. The matrix $K$ is called the gain matrix and is estimated using the three parameters $\epsilon^2$, $D_h$ and $D_z$. The three parameters act similarly to cut-off wavelengths of a low pass filter, being proportional to the confidence in the observation data over the background field data ($\epsilon^2$) and the size of the horizontal ($D_h$) and vertical ($D_z$) "influence area" of each station. Given a certain distribution of station points, the three parameters are thus chosen by first evaluating the *integral data influence* (IDI) field, meaning the $x^a$ values obtained when all background field values are set to 0 and all station values are set to 1. Calibration of the parameters aims at creating an IDI field as uniform and as close to 1 as possible. Given that the number of available stations has changed throughout the series (from 25 to 371 and from 119 to 386 for temperature and precipitation stations respectively), the parameters needed for the calculation of the $K$ matrix were chosen on a year-by-year basis.

The described interpolation method was applied to both temperature and precipitation data, using two different background fields and calculating two different sets of parameters for each year. For the temperature interpolation, the reanalysis set ERA-40 (Kållberg et al., 2004) published by the European Centre for Medium Range Weather Forecast (ECMWF) for the 1957-2001 period and, for the following period, ECMWF analyses' set were used as background fields. For the precipitation no external background field was used, instead calculating a pseudo-background field from the station data itself via linear function of the spatial coordinates by performing a least-square minimization (Uboldi et al., 2008). Furthermore, the interpolation of the precipitation data was rendered two-dimensional by assigning the same height to each station point (temperature interpolation was instead three-dimensional) due to considerations about the high variability of conditions inside each cell, particularly in the alpine area.

## Appendix B: Trend analysis on seasonal precipitation and temperature values

As discussed in Section 3.3.1, trend analysis is conducted on monthly precipitation and maximum/minimum temperatures on both an annual and seasonal scale. Seasons are defined as the three month periods December-January-February (Winter),

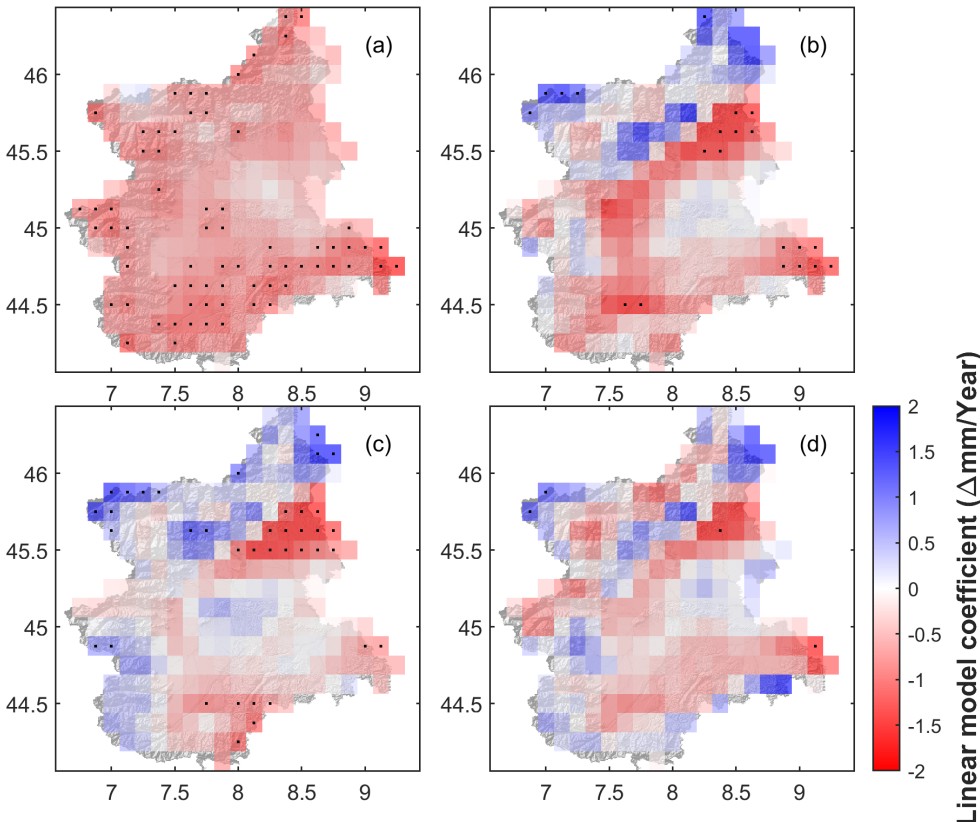

**Figure B1.** Seasonal trend analysis on monthly precipitation values. **(a)** Winter, **(b)** Spring, **(c)** Summer, **(d)** Autumn. Cells containing a dot denote significant trends at 5% significance.

March-April-May (Spring), June-July-August (Summer), September-October-November (Fall). Results for the seasonal trend analysis of monthly precipitations are reported in Figure B1; results for the seasonal trend analysis of monthly maximum temperatures are reported in Figure B2; results for the seasonal trend analysis of monthly maximum temperatures are reported
in Figure B3

**Appendix C: Spatial distribution of drought run characteristics**

As discussed in section 3.3.1, the spatial distribution of drought run characteristics is analyzed, particularly in its correlation with altitude/terrain roughness. Maps of the spatial distribution of drought characteristics are shown for SPI-3 and SPEI-3 in Figure C1, and for SPI-12 and SPEI-12 in Figure C2.



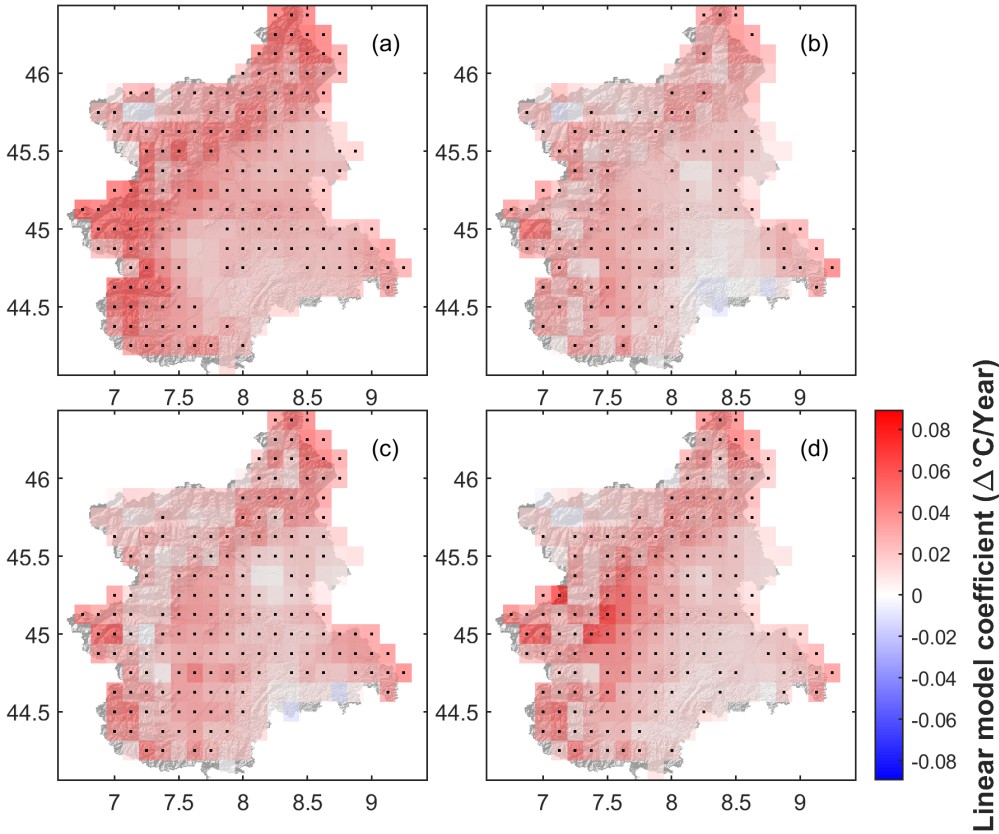

**Figure B2.** Seasonal trend analysis on monthly maximum temperature values. **(a)** Winter, **(b)** Spring, **(c)** Summer, **(d)** Autumn. Cells containing a dot denote significant trends at 5% significance.

**Appendix D:  Drought event analysis**

As discussed in Section 3.4, drought event analysis is conducted on SPI and SPEI at 3 and 122 month scales. The results for SPI-3 and SPEI-3 are shown in Figure D1, the results for SPI-12 are shown in Figure D2.

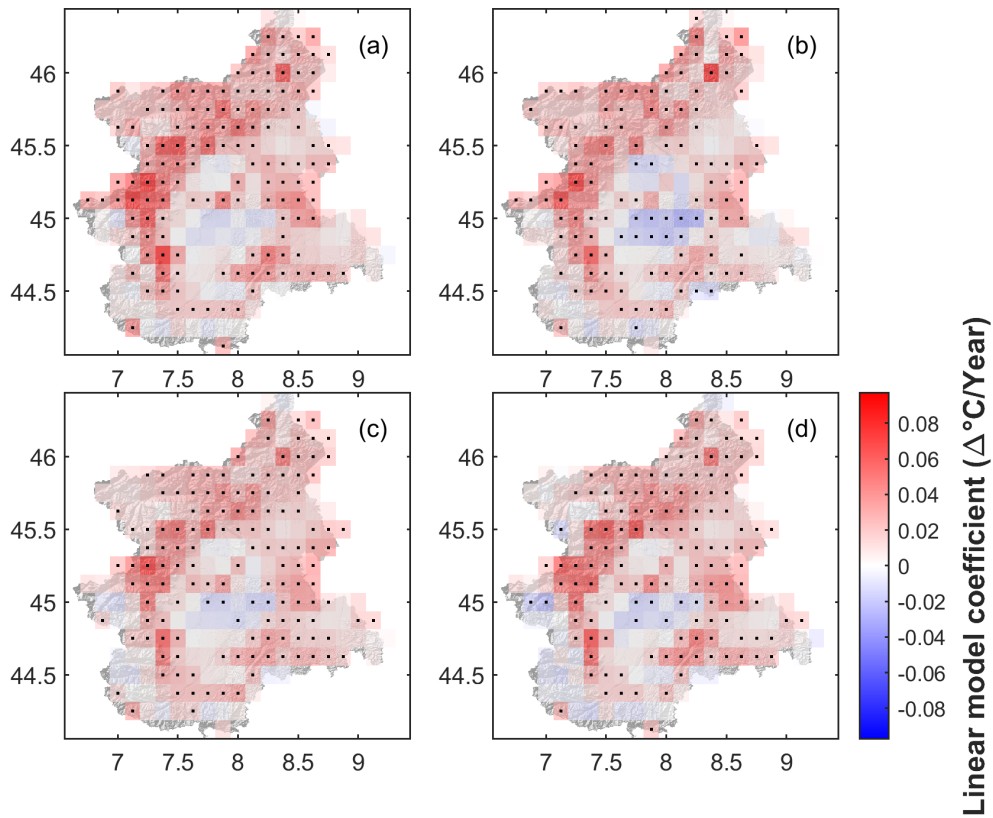

**Figure B3.** Seasonal trend analysis on monthly minimum temperature values. **(a)** Winter, **(b)** Spring, **(c)** Summer, **(d)** Autumn. Cells containing a dot denote significant trends at 5% significance.

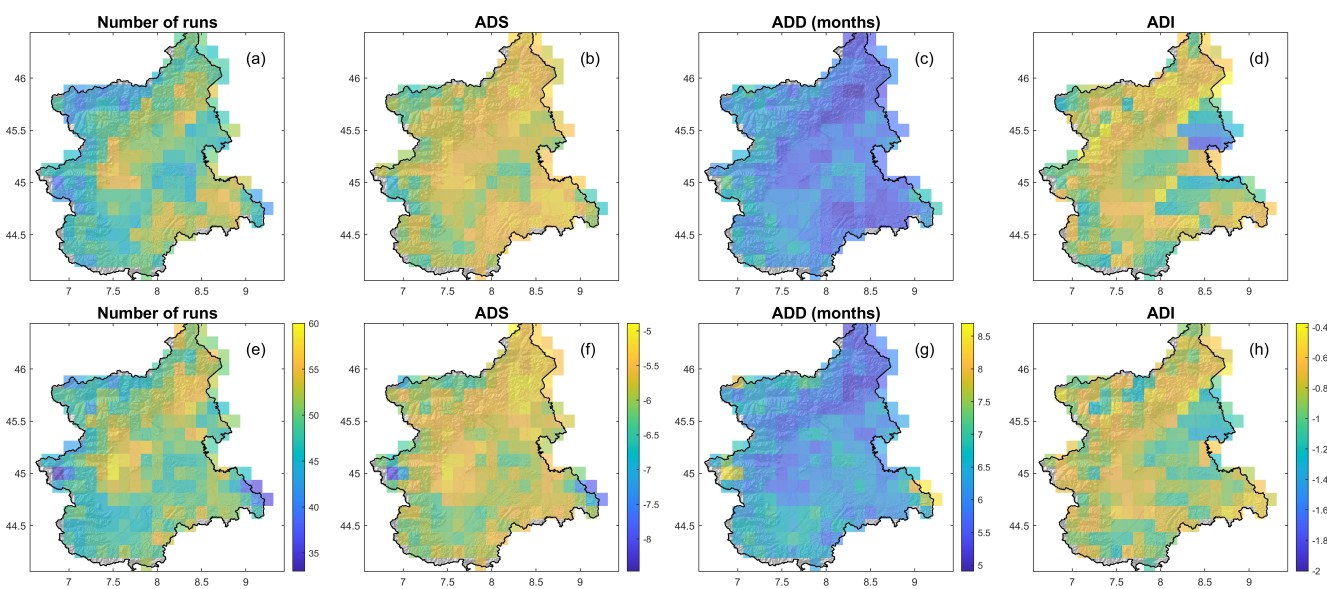

**Figure C1.** Spatial distribution of drought run characteristics at 3 month time scale. **(a-d)** SPI-3. **(e-h)** SPEI-3.

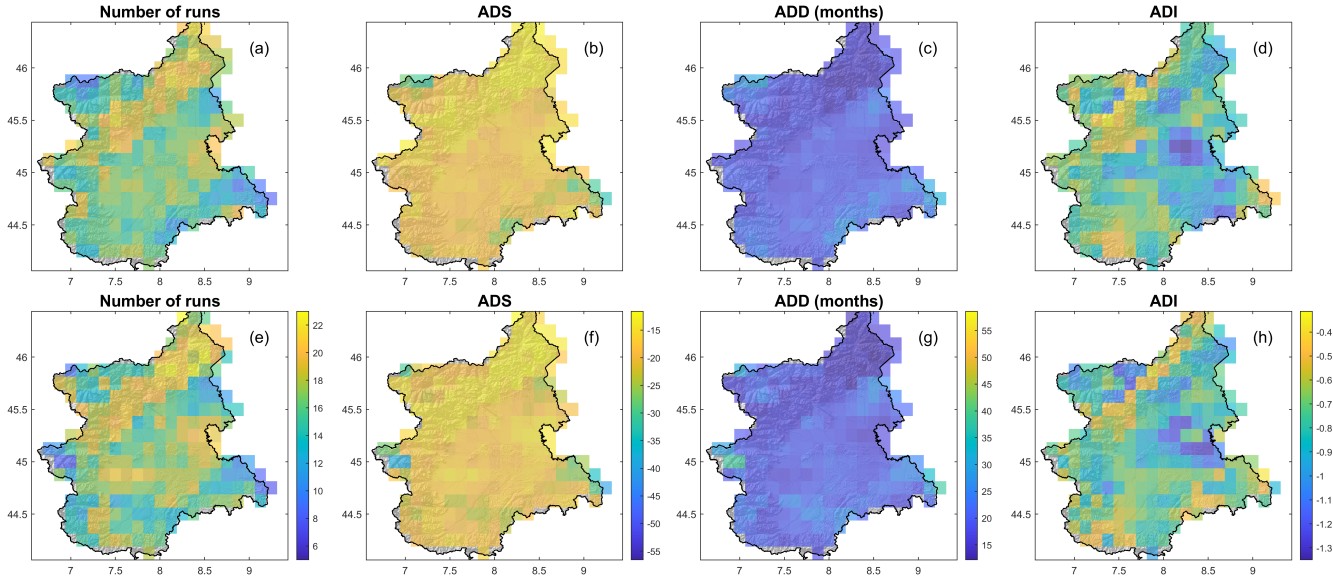

**Figure C2.** Spatial distribution of drought run characteristics at 12 month time scale. **(a-d)** SPI-12. **(e-h)** SPEI-12.



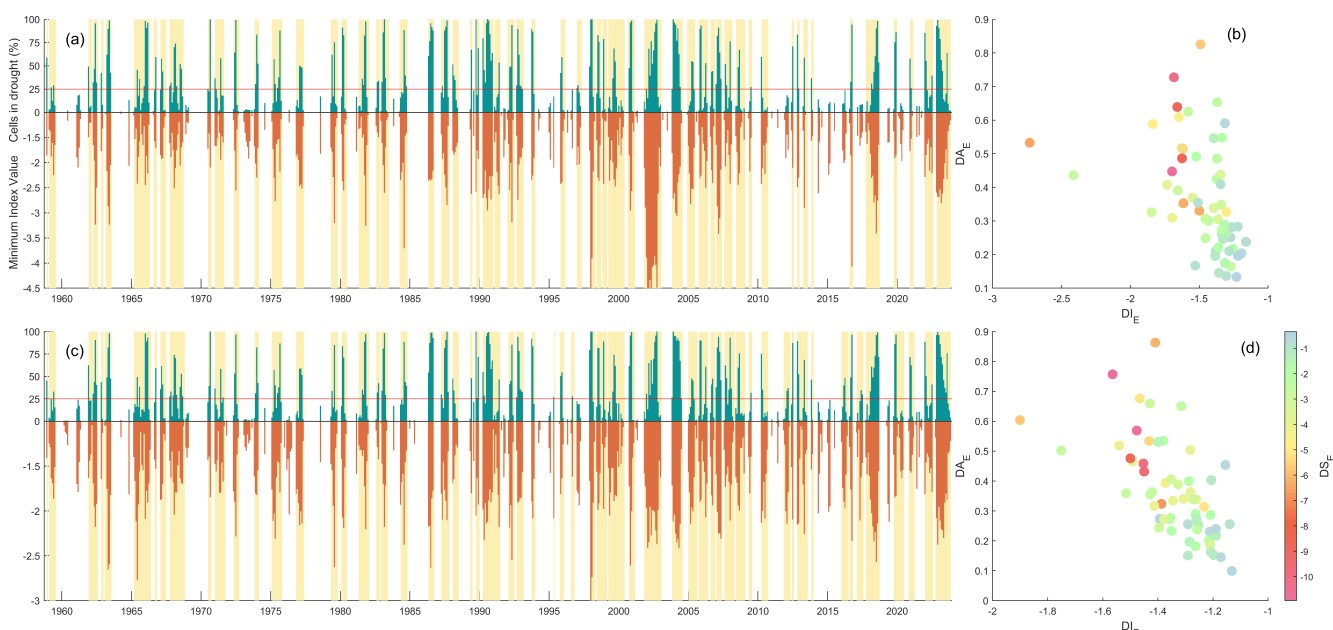

**Figure D1.** Drought event analysis conducted on the indices at 3 month scale. **(a)** Series of percentage of cells in drought condition (below the -1 threshold) and the minimum index value in the domain for SPI-3. **(b)** Drought event characteristics for SPI-3. **(c)** Series of percentage of cells in drought condition (below the -1 threshold) and the minimum index value in the domain for SPEI-3. **(d)** Drought event characteristics for SPEI-3.

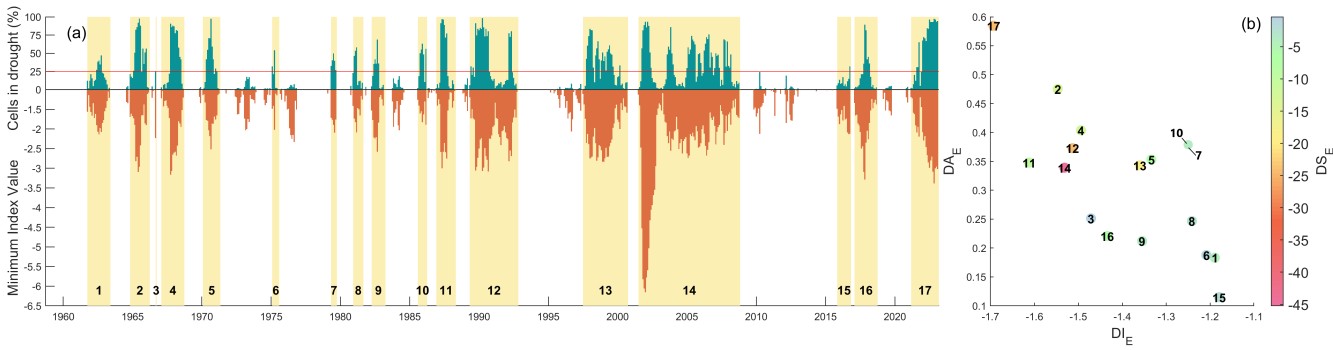

**Figure D2.** Drought event analysis conducted on SPI-12. **(a)** Series of percentage of cells in drought condition (below the -1 threshold) and the minimum index value in the domain. Each event is highlighted in yellow and labeled. **(b)** Drought event characteristics.





*Author contributions.* M.E., S.T., A.V. and R.R. contributed to the design of the research, to the analysis of the results and to the writing of the manuscript. M.E. carried out the analysis.

*Competing interests.* The authors declare no competing interests.

*Acknowledgements.* This study was carried out within the RETURN Extended Partnership and received funding from the European Union Next-GenerationEU (National Recovery and Resilience Plan – NRRP, Mission 4, Component 2, Investment 1.3 – D.D. 1243 2/8/2022, PE0000005).





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
