# Peer review of "60-years drought analysis of meteorological data in the western Poriver Basin"

_Hydrology and Earth System Sciences, 2023_

## Author Response (AR1)

We present here a point-by-point response to the comments made by Reviewer #1 and #2. Other than the changes described in the following, we have performed minor editing on the manuscript (visible in the track-changes file), rephrasing some paragraphs and correcting minor mistakes.

**Response to reviewer #1**

We thank Reviewer #1 for the detailed analysis and the precise comments about our methodology and presentation/discussion of the results. We respond here to the more specific concerns raised and how they are addressed in the revised manuscript. In the following, Reviewer #1 comments are in plain text, while our comments are in italics.

- The findings are not contrasted with other studies and the novelty for hydrology and earth system sciences is unclear.

*The Discussion section has been expanded by including a comparison between the presented study and similar ones found in the literature, indicating the novelty of our study and the originality of its results (see Lines 485-501 of the revised manuscript). In particular, we state that "[our] type of analysis is in common with a growing body of literature focused on the elevation effects on drought characteristics" (Line 488-489 of the revised manuscript), and that "our study shows that mean elevation, although certainly a variable to be considered, shouldn't be the only topographic variable taken into account" (Line 495-496 of the revised manuscript) given that "[i]n our analysis, using [a] different classification leads to stronger correlations between drought characteristics and topographical characteristics" (Line 499-500 of the revised manuscript).*

- Trend attribution is not investigated or discussed.

*Although the attribution of the identified trends is surely an interesting issue, it is out of the scope of the present paper, which is focused on trend detection for both drought indices and drought event characteristics. We have however now mentioned the attribution issue in the discussion section, as a possible further development of the research. The following lines have been added (Lines 502-504 of the revised manuscript): "Although strong correlations between drought trends and the mean elevation and ruggedness of the terrain are found, attribution of these results to physical phenomena is not straightforward. The presented methodology doesn't focus on this aspect and, given the complexity of the involved phenomena, attribution is outside the scope of our study".*

• The first paragraph of the Introduction is too general and not directly associated with the aim of the study. It would read better if presented more concisely.

*The paragraph has been rewritten and made more concise, see Lines 19-23 of the revised manuscript.*

• Line 36 needs citations for "changing patterns of meteorological droughts"; and also for "an increase in drought occurrence in the area has been detected".

*The phrase "changing patterns of meteorological drought" was intended as a summary of the aim of the cited studies (i.e. trend detection, changes in drought characteristics...); in order to avoid confusion, it has been removed. The paragraph has been rearranged to make clearer that the "increase in drought occurrence" is a summary of the results in the cited literature, see Lines 26-37 of the revised manuscript.*

• Lines 36-43 are unclear. When and where are such changes detected? How do the "reported changes differ significantly"? How consistent are the following results: "studies considering both precipitation and temperature (Hanel et al., 2018; Falzoi et al., 2019; Arpa Piemonte and Regione Piemonte, 2020b; Vogel et al.,

2021; Baronetti et al., 2022) have found more consistent results"? Why is "the rise in evaporation as a main factor in drought increase"?

•

*Lines 31-37 of the revised manuscript have been changed to address the Reviewer #1 comments. In particular, Lines 36-43 of the old manuscript have been changed in order to convey the area and the time period to which the cited results apply ("Overall, these studies have found an increase in meteorological drought occurrence in north-west Italy, particularly after the 1970s"). Furthermore, the cited significant difference in the reported precipitation change refers to the presence of either summer or winter precipitation decrease, and so this has been made more explicit ("On the other hand, studies considering temperature..."). The consistency of the results related to temperature increase (and thus to evaporative demand) is now mentioned ("consistently shown rising temperatures, and thus a rise in evaporative demand, to be a main factor in drought increase"). Finally, an erroneous citation (Baronetti et al., 2022 instead of Baronetti et al., 2020) has been corrected.*

•       Lines 48-49: What are the results conflicting and what is the consensus?
*The conflicting results are the presence of a relation between the latitude and the temperature trend, while the cited consensus is the presence of enhanced warming at higher altitudes; Lines 48-49 (old manuscript) have been modified in order to make them clearer: "In general, despite conflicting results regarding the presence of an elevation effect on warming rates and the lack of adequate climate data for mountainous regions, a consensus on enhanced warming rates at higher altitudes emerges" (Lines 42-44 of the revised manuscript).*

•       Lines 51-54: When and where did these studies investigate? Does it also cover the study area of the current manuscript?

*The meta-analysis mentioned at Lines 52 (Pepin et al., 2022) is based on studies from both station data and gridded datasets related to various mountain ranges (including the Alps); the study period also varies widely, ranging from the early 1950s to the end of the 2010s, with varying study periods lengths. Lines 52-53 of the old manuscript have been revised to mention this information: "A comprehensive meta-analysis of both in-situ studies of precipitation data from mountainous regions (including the Alps) and of global gridded databases from the early 1950s to the late 2010s reported a relative decrease in precipitation compared to lowlands, although without high confidence" (Lines 46-48 of the revised manuscript). The study of Giorgi et al. (2016) focuses on the western part of the Alpine range and compares climate models (in the present, near-future and future) at different scales using high resolution observation data (for the 1975-2004 period) as a baseline. The fact that this study does include the present study area is mentioned in the revised manuscript: "Furthermore, analyses such as Giorgi et al. (2016) have shown the importance of spatial resolution in understanding these processes in topographically complex regions, reporting that increases in summer precipitation in higher elevation areas of the Alpine range could only be detected by high resolution regional climate models and observed by high resolution observation networks." (Lines 47-51) of the revised manuscript).*

•       Lines 58-60: Why did the authors choose the present study area? For example, does it have longer or more extensive observed data sets to analyze the effect of elevation on meteorological trends? Please specify that only meteorological droughts are investigated. It is also proposed to contrast the drought conditions in northern Italy to those in northwestern Italy, but the manuscript only presents for northwestern Italy.

*We opted for North-Western Italy due to our familiarity with the region (given our work in Turin) and the significant institutional and public concern regarding the onset and characteristics of droughts, particularly following the unprecedented drought of 2022 in the area. Lines 58-60 of the old manuscript have been changed to address Reviewer #1's comments: "Understanding the possible effects of topographically related phenomena on drought conditions is thus of particular interest in an area such as the western Po river basin, which comprises both wide plains and high mountains. Despite the presence, as detailed above, of studies on drought in the chosen region, these lacked either the needed spatial resolution or focus to evaluate possible effects of terrain characteristics on drought conditions." (Lines 52-55 of the revised manuscript).*

*Furthermore, we propose to change the title to "60-years drought analysis of meteorological data in the western Po river basin" in order to convey the meteorological drought focus of the article.*

• Figure 1. What are the elements shown in the larger map (colors, lines, names)? Is it elevation, land cover, rivers, roads? Every element should be either described or removed from the map. The river network should also be included, as many readers may not know the extent of the Po river. Roads and region names that are not relevant to the study should be removed. Also, the same projection should used in all maps, so that the shape of the study area (and the latitude-to-longitude ratio) on the larger and smaller maps are the same. Similar reasoning applies to the other figures.

*Figure 1 has been modified according to Reviewer #1's suggestions.*

• Line 71-72. Do you mean "bordered by France on the west and south-west"? And "two other Italian regions (…) on the east and south-east?

*The reported typo has been corrected, see Lines 69-71 of the revised manuscript.*

• The latitudes described in lines 76-77 and 96 are not needed and can be inferred from Fig. 1.

*The latitude/longitudes values have been omitted, see Lines 69-72 of the revised manuscript.*

• A description of the data projection in Line 96 is not needed. The total number of grids described in lines 96 and 106 are also not needed because the spatial resolution of the data is already provided.

*The number of grid points cited at Line 96 of the old manuscript has been removed (see Line 93 of the revised manuscript). The number cited at Line 106 of the old manuscript has not been omitted as it refers to the number of actual series studied, meaning points in the dataset actually falling inside of the studied domain.*

• Please provide a reference to the data set (Line 95) and the interpolation method (Line 97).

*The references to the dataset and to the interpolation method used have been added, see Lines 91-2 and 95 of the revised manuscript.*

• Section 2.2. Why are these data explored and how do the results change if other data sets are analyzed? Do the data explored have a dense gauging network? How many gauges does it include? It is important to provide specific data details due to the small study area and the high result sensitivity to different data sources (cited in the Introduction).

*Our choice of dataset is functional for the purposes of our study, because of its spatial resolution and its observation-based nature. As detailed in Appendix A, the interpolation method used for the dataset doesn't add or remove elevation trends and is thus suitable for studying the relations between meteorological variables and terrain characteristics. Furthermore, the dataset has a much higher number of gauging stations in the region compared to other datasets, with hundreds of stations in the area (see https://doi.org/10.5194/nhess-13-1457-2013). More details on the distribution of stations over the domain and their number, as well on the density of the chosen dataset compared to other available ones have been added in the revised manuscript: "The data used in the interpolation method is provided by a dense gauging network (roughly 200 stations) covering both low and high altitude areas, providing a much higher number of stations than other available datasets for the area (Turco et al., 2013)." (Lines 99-101 of the revised manuscript).*

•	Figure 2. Is "Terrain roughness" the same as "Elevation standard deviation"? Also, is it "terrain roughness" or "terrain ruggedness"? It would be nice to standardize throughout the text. Please clarify the units of "Terrain roughness". What does F and F(x) refer to?

*Terrain ruggedness is calculated as the standard deviation of the elevation inside a cell, and as such has a unit of meters; to avoid confusion, the term "elevation standard deviation" is not used outside of the explanation at Lines 110-111 of the revised manuscript. Furthermore, the use of the term terrain ruggedness (sometimes cited wrongly as "terrain roughness" in the old manuscript) has been standardized throughout the manuscript. Finally, F(x) referred to the empirical cumulative distribution function of terrain ruggedness values, and this has been made explicit in the revised figure.*

•	Lines 113-116. What is the definition of terrain ruggedness (its concept) and how is it calculated? Please provide precise details in the main text. Also, why did the authors choose the present metric and how does it compare with other metrics (e.g., terrain slope, amplitude, other terrain ruggedness indices)?

*A definition of terrain ruggedness and citations for how it can be calculated are now provided in the revised manuscript: "The terrain ruggedness (also known as surface roughness or topographic hetereogenity) is defined as the "deviations in the direction of the normal vector of a real surface from its ideal or intended form" (Whitehouse, 1994), meaning the irregularity of a landscape" (Lines 108-110 of the revised manuscript). The choice of terrain ruggedness as a variable is motivated by a need to differentiate areas with distinct terrain characteristics (plains, hills and mountains) which would have been grouped together if classified through elevation bands. We acknowledge that other variables could have been used, such as those proposed by Reviewer #1, but the impact of a similar classification would not change the results. As an example, the following Figure shows the results of a classification based on mean slope inside a cell, obtained from the same DEM as the terrain ruggedness (this figure can be compared with Figure 2 of the revised manuscript):*

[Figure]

•	Line 113: "height". Do you mean "elevation"?

*The use of elevation instead of height has been standardized throughout the revised manuscript.*

•	Lines 116-118 are a repetition of the previous section. Either make it more concise or remove the sentence.

*The lines have been removed.*

•        Lines 120-123. A description of the results is not needed here. Can remove these sentences.

*The Lines have been removed in the revised version of the manuscript.*

•        Lines 124-127. I could not find the results compared with the K3 Mountain classification. How did the authors find that the classification is "quite satisfactory"? How satisfactory is it?

*The classification has been deemed satisfactory as it correctly distinguishes between the hilly region at the center of the domain and the surrounding plains, as well as correctly identifying mountainous areas, as can be seen in the following figure comparing the terrain ruggedness classification and the K3 Mountain classification.*

[Figure]

•        Table 1. These thresholds were not used or discussed in the manuscript. Thus, the table could be removed.

*Table 1 has been removed, and the only reference threshold (SPI < -1) has been maintained in the text: "a series of consecutive months under a certain threshold (-1, corresponding to a moderately dry condition in the SPI classification)" (Line 193-194 of the revised manuscript).*

•        Lines 157-159. These sentences are unclear.

*Lines 157-159 of the old manuscript report the way in which the parameter of the gamma distribution have been estimated and the normal inverse function has been calculated for the current study. This was made explicit because these two steps are usually performed via linear approximations given in the literature rather than with dedicated software functions.*

•        Line 164. Is it potential evaporation?

*$ET_0$, in the context of SPEI calculation, refers to the "Reference Evapotranspiration", meaning the maximum evapotranspiration from a reference well-watered reference "alpha-alpha grass" surface (see https://doi.org/10.1016/j.agwat.2020.106043 for the differences with potential evapotranspiration, and https://doi.org/10.1002/joc.3887 for the reasons for its use in the SPEI calculation).*

•	Line 184. How were the series deseasonalized? With "seasonal precipitation series", do you mean the series for each season, or the series with the seasonal time scale?

*Deseasonalization was done using a function in the Climate Data Toolbox (https://doi.org/10.1029/2019GC008392), which removes the mean of the detrended series for each month. The missing citation for the method has been added: "Furthermore, deseasonalization is performed by subtracting the mean of the detrended temperature series for each month using the Climate Data Toolbox (Greene et al., 2019)." (Lines 177-179) of the revised manuscript. Seasonal precipitation series were meant as the series of cumulative precipitation values for different seasons; this definition is now provided at Lines 175-177 of the revised manuscript: "Seasonal values are defined as the cumulative precipitation values and the mean temperature values over the four three-months periods December-January-February (Winter), March-April-May (Spring), June-July-August (Summer), September-October-November (Fall)".*

•	Figure 3. I could not find DDr and DSr in the figure.

*Figure 3 has been changed and now includes DDr and DSr.*

•	Lines 200-202. It would be nice to clarify the differences between Fig. 3a and 3b. Also, please define precisely when the drought ends in each case. How sensitive are the results for different thresholds in defining when the drought ends?

*A more detailed description of the method used for the definition of drought runs has been added to the revised manuscript: "The differences between the use of a single threshold and the inclusion of the run onset and offset are shown in Figure 3: the method used in the present study considers two periods of months with index value under the -1 threshold as part of the same run if the drought index remains lower than 0 between them; in any case, a drought run ends if the drought index becomes positive." (Lines 196-199 of the revised manuscript). The use of a different threshold for the definition of the end of local droughts were considered, both using a simple –1 threshold (as in Figure 3 (a) of the revised manuscript) and the inclusion of months with negative index values only after a month under –1 threshold. Results for this latter method differ very slightly from the ones presented in our study. As an example, the Pearson correlation values between the terrain ruggedness and the change in drought run characteristics obtained with this method are reported in the following table (these values can be compared with those of Table 3 of the revised manuscript):*

|  | Δ Number of runs | | Δ Mean DS$_R$ | | Δ Mean DD$_R$ | | Δ Mean DI$_R$ | |
|---|---|---|---|---|---|---|---|---|
|  | C | p-value | C | p-value | C | p-value | C | p-value |
| SPI-3 | 0.18 | 5.22E-03 | 0.43 | 1.62E-11 | -0.35 | 7.73E-08 | 0.40 | 4.09E-10 |
| SPEI-3 | 0.23 | 5.57E-04 | 0.37 | 1.26E-08 | -0.33 | 3.10E-07 | 0.15 | 2.37E-02 |
| SPI-12 | -0.35 | 5.37E-08 | 0.24 | 2.09E-04 | -0.26 | 9.66E-05 | 0.17 | 9.38E-03 |
| SPEI-12 | -0.30 | 6.35E-06 | 0.32 | 1.13E-06 | -0.29 | 7.84E-06 | 0.32 | 8.67E-07 |

*Results obtained using a single threshold differ more significantly, especially regarding the mean drought characteristics value, but the overall results still support the conclusion presented in our study (relative values of drought characteristics between SPI and SPEI, change of drought characteristics over time and correlation between drought characteristics and terrain characteristics). As an example, the Pearson correlation values between the terrain ruggedness and the change in drought run characteristics obtained with a simple threshold method are reported in the following table (these values can be compared with those of Table 3 of the revised manuscript):*

|  | Δ Number of runs | | Δ Mean DS$_R$ | | Δ Mean DD$_R$ | | Δ Mean DI$_R$ | |
|---|---|---|---|---|---|---|---|---|
|  | C | p-value | C | p-value | C | p-value | C | p-value |
| SPI-3 | 0.20 | 2.89E-03 | 0.59 | 5.09E-23 | -0.57 | 1.24E-20 | 0.64 | 9.01E-28 |

| | | | | | | | | |
|---|---|---|---|---|---|---|---|---|
| SPEI-3 | 0.14 | 3.75E-02 | 0.52 | 7.75E-17 | -0.48 | 2.03E-14 | 0.54 | 1.99E-18 |
| SPI-12 | -0.36 | 2.52E-08 | 0.09 | 1.62E-01 | -0.08 | 2.30E-01 | 0.16 | 1.88E-02 |
| SPEI-12 | -0.28 | 1.77E-05 | 0.30 | 5.82E-06 | -0.25 | 1.88E-04 | 0.35 | 5.07E-08 |

• Line 213. "discarded". Do you mean "analyzed"?

*The line referred to the fact that the criteria of a minimum duration of three weeks for drought events, as proposed in the literature, is always met in this study given the use of monthly data. Lines 213-214 of the revised manuscript has been changed to better convey the intended meaning: "the minimum duration threshold of 3 weeks, used in the cited papers, is always met as monthly data is used in this analysis".*

• Sections 2.4.4 and 2.4.5. I find it hard to distinguish between "drought run", "drought event", "drought episode" in the Methodology and the Results sections. It might be clearer to refer to droughts in a single cell as "local droughts" and in multiple cells as "regional droughts", or some other term related to the spatial differences. Please standardize throughout the manuscript.

*The use of the terms "drought run" and "drought event" has been changed in the revised manuscript following Reviewer #1's comment, now referring to them as "local droughts" and "region-wide drought events" to better differentiate between the two. Furthermore, the pedex R previously used to denote the drought run characateristics has now been changed to L in order to better reflect the local level of the analysis.*

• Figure 4. (a) Should also be presented and discussed in relative terms (units of % per year relative to the long-term precipitation). The negative values in the colorbar of (b) and (c) should have a smooth transition from zero, symmetric to the positive values. The colorbar in (a) is ok. Please label the numbers in the x and y axis (∘N and ∘E) and use the same projection as Figure 1.

*Figure 4 has been changed according to Reviewer #1's comments. The change in terms of relative precipitation change is now discussed at Lines 261 and 262-263 of the revised manuscript.*

• Line 251-252. Figure 4 does not present seasonal trends.
*References to the Figure 4 and supplementary Figure B1 have been revised in order to avoid confusion, see Lines 255-256 of the revised manuscript.*

• Line 255-257. Do the authors mean that precipitation trends are not the same in the entire area?

*When trend analysis is performed on the spatial average of the precipitation over the whole study-domain no significant trend is found. Lines 255-257 of the old manuscript were meant to convey this, and they have been revised in order to be clearer: "Still, spatially averaged precipitation over the whole region does not show a statistically significant decrease either at the annual or seasonal scale." (Lines 262-265 of the revised manuscript).*

• Correlation values should always specify their associated p-values and which correlation method is used (Pearson, Spearman). If Pearson correlation is used, are its assumptions met? These should be clarified in Lines 260, 265, and throughout the manuscript.

*Correlation values reported in the manuscript have been calculated using Pearson's methods, but Spearman's values have also been calculated following Reviewer #1's comments. In the following figures the scatter plots between drought run characteristics and mean elevation/terrain ruggedness and between drought run characteristics change from the first to the second half of the study period and mean elevation/terrain ruggedness are reported:*

*Drought run characteristics – Mean elevation (values reported in Table 1 of the revised manuscript)*

[Figure]

*Drought run characteristics – Terrain ruggedness (values reported in Table 2 of the revised manuscript)*

[Figure]

*Drought run characteristics change – Mean elevation (values discussed but not reported in the manuscript)*

[Figure]

*Drought run characteristics change – Terrain ruggedness (values reported in Table 3 of the revised manuscript)*

[Figure]

*As can be seen from the previous figures, the reported Pearson's and Spearman's correlation values are quite close, with Spearman's values usually higher and more significant. Given that the scatter plots show linear correlations, and to be more conservative in the reported results, Pearson's values are the ones reported in the revised manuscript. The fact that both correlation values have been calculated and that only one type is reported is now mentioned in the revised manuscript at Lines 234-237. Finally, by checking the correlation values an error in the calculation of the mean drought intensity value—and thus its correlation value with either mean elevation or terrain ruggedness — was found: the error has been corrected and Tables 1 and 2, as well as their discussion at Lines 347-365 of the revised manuscript, and Figures C1 and C2 (which reported the wrong mean drought intensity values) have been amended.*

• Line 279-281. Why are these associated with soil moisture and groundwater? Please provide citations or the results of an analysis.

*The correlation between SPI/SPEI at different time scales with different water resources is usually accepted in the literature as a property of the indices based on the propagation of drought in the hydrological cycle (see for example the Standardized Precipitation Index User Guide issued by the WMO, https://library.wmo.int/records/item/39629-standardized-precipitation-index-user-guide). Although this correlation has been studied and somewhat confirmed in the literature (e.g. https://doi.org/10.5194/hess-9-523-2005), our reference to it is based just on the generally accepted operative use of the drought index. Thus, to avoid confusion with an actual water resources/drought index correlation analysis, we prefer to leave it without citation. We have however reformulated the sentence as follows: "Furthermore, trend analysis on indices at the shorter 3 month time scale and the longer 12 months time scale indicates, respectively, how drought conditions might have evolved over smaller time scales, closer to the response time of soil moisture conditions to meteorological conditions, and over larger time scales, closer to the response time of water reservoirs and groundwater levels to meteorological conditions." (Lines 288-291 of the revised manuscript)*

• Lines 283-285. These sentences are unclear, please revise them. What are "worse conditions"?

*Lines 283-285 of the old manuscript have been revised: "Trend analysis on SPI-3 and SPI-12 values shows results that mostly agree with the trends in annual precipitation, as a majority of cells reports both significant negative trends in annual precipitation values and in index values (and thus a tendency towards dryer conditions)." (Lines 293-294 of the revised manuscript). "Worse conditions" referred to a dryer overall climate, represented by lower SPI values over time; as such, "worse conditions" have been changed to "dryer conditions" in the revised manuscript, see Lines 294-295.*

• Figure 5. (a) Is precipitation units in mm per month? This figure is not discussed in the manuscript. Either discuss it or remove it.

*The figure was intended to give a visual representation of the type of data analyzed in the following sections; since we now feel that it is not needed and that it could lead to confusion, as also noted by Reviewer #2, it has been removed.*

• Lines 290-293. How relevant is the magnitude of the SPI and SPEI trends?

*Drought index trends indicate, if negative, a downward shift of precipitation/precipitation minus reference evapotranspiration (for SPI and SPEI respectively) values, meaning that over time it is more likely that a given month will have lower-than-average values and thus be in drought conditions. The magnitude of the trend is thus related to how much the average conditions are shifting towards the lower part of the meteorological values' distribution: for example, the mean for SPEI-12 is 6% change in a decade (see Figure 5 of the revised manuscript), comparable with the change in percentage of the average annual precipitation (Figure 4 of the revised manuscript). Still, as we show in Sections 3.3 and 3.4, this influences various drought characteristics leading to longer and more severe drought periods, as well as drought conditions influencing wider portions of the region at the same time.*

• Figure 6. (e) and (f) should avoid differentiating the variables by blue and red colors because it creates some confusion with (a) – (d). Also, what exactly is the unit Δindex? Why are the trends presented in month units here but in year units in Fig. 8?

*The colors denoting the different indices have been changed in order to avoid confusion. The Δindex unit is represents the change in the different indices (SPI-3,SPI-12,SPEI-3,SPEI-12) given their standardized nature and their lack of a unit measure. Finally, in order to avoid inconsistencies, the trends are now presented in yearly change rather than monthly change.*

• Table 2. Are "Number of runs" the number of drought events? Is C the correlation coefficient? Why is it that only one variable has units?

*In Table 2 "Number of runs" refers to the number of local droughts in a cell: the mean severity, duration and intensity are then $DS_L$, $DD_L$ and $DI_L$ respectively. Of these latter values only $DD_L$ has a units as it is measured in months, while the other two are a sum and a median of a standardized value lacking a unit measure. To avoid confusion, and since it is not needed to present a correlation coefficient, the unit measure of $DD_L$ has been omitted. Finally, C is the correlation value, and this has been made explicit in the table caption.*

• Figure 8. This figure is hard to understand and is also not much discussed in the manuscript. Red and blue colors are used in the other figures to differentiate between increasing or decreasing trends, but here denote different variables, creating some confusion.

*The colors used to indicate the different drought indices have been changed in order to avoid confusion.*

• Figure 9. (a) Do the negative and positive y values represent different variables? If so, this should be clarified by using two different y axis and by describing in the figure caption.

*Figure 9 has been changed according to Reviewer #1's comment.*

• Line 422. What does "worse" refer to here?

*"Worse" referred to drought event characteristics (severity and duration) becoming more severe. The phrase has been changed to avoid confusion: "This seems to confirm that the shift towards worse region-wide drought conditions (higher severity and longer duration) is more evident at longer time scales, and that this shift is mainly caused by a change in precipitation patterns." (Lines 436-438 of the revised manuscript).*

**Response to reviewer #2**

We thank Reviewer #2 for the detailed analysis and the precise comments about our methodology and presentation/discussion of the results. We respond here to the more specific concerns raised and how they are addressed in the revised manuscript. In the following, Reviewer #2 comments are in plain text, while our comments are in italics.

- If the aim is indeed to investigate the correlation between drought trends and elevation/ruggedness this should be better reflected in the methods and results, since as it is now only a small part of the results takes into account the elevation/ruggedness and the rest just focuses on the trends in droughts.

  *The aim is not only to investigate the correlation mentioned by the Reviewer but also to present drought characteristics and detect changes in time. However, the relationship between drought changes and elevation is central. Therefore, the Method section of the revised manuscript now focuses more on the different classification of the domain based on mean elevation/terrain ruggedness (se Lines 105-119 of the revised manuscript); furthermore, following Reviewer #1's suggestion, an additional section concerning the way in which correlation values are calculated has been added (see Lines 233-237 of the revised manuscript). Furthermore, the Discussion section of the manuscript has been revised, adding a comparison between the literature already analyzing drought-elevation relations and how our study adds to this field, and also providing a further discussion of the results obtained (see Lines 485-501 of the revised manuscript).*

- The first part of the introduction is very general and not very relevant.

*The first paragraph has been made more concise: "Drought is considered to be one of the main natural disasters, with widespread effects affecting large portions of the world's population (Wallemacq et al., 2015) and causing severe financial losses (García-León et al., 2021) and ecosystem impacts (Crausbay et al., 2020). Drought also has both short- and long-term effects on water availability (IDMP, 2022), which are relevant when considering the global increase in water demand in the last 100 years and the predicted challenges in meeting that demand in the future (Unesco, 2018; Wada et al., 2016; Burek et al., 2016)" (Lines 19-23 of the revised manuscript).*

- In lines 36 to 38 the authors state that the fact that drought occurrence increases is contradictory to the finding that recent droughts are not exceptional. However, this does not necessarily contradict each other, droughts can occur more frequently, even if the individual droughts are not more exceptional than previous droughts.

*We agree with Reviewer #2 consideration, and we have modified Lines 36 to 38 of the old manuscript in order to avoid confusion about their intended meaning: "Overall, these studies have found an increase in meteorological drought occurrence in North-West Italy, particularly after the 1970s, even when recent drought events have not been found to be exceptional when compared to historical records" (Lines 31-32 of the revised manuscript).*

- In general, the introduction describes a lot of research on drought and its relation to orography that has already been done in Italy. It is not very clear to me which research gap the authors aim to address with this study and how this will contribute to an improved understanding of drought.

*Drought has been studied in the area by focusing on the trends in drought indices, and no studies have investigated possible link between drought and terrain characteristics. We feel that such studies are important, given the growing literature focused on the elevation dependent warming/precipitation-change effects under climate change conditions. Furthermore, while some studies have considered drought-elevation relations in other parts of the world (China https://doi.org/10.1038/s41598-020-71295-1, Iran*

*https://doi.org/10.1007/s00704-020-03386-y, India https://doi.org/10.1016/j.atmosres.2023.106824 and the Canary Islands https://doi.org/10.1038/s41612-023-00358-7),* *our study area presents topographical characteristics that lead to evidence of more complex interactions between terrain characteristics and wetting/drying trends, as the ruggedness of the terrain is better correlated than elevation to the observed trends/changes.*

- In section 2.2 the authors describe that the data set they use is a gridded dataset, based on the interpolation of station data. How are these results affected by the interpolation method used? Why not analyse the station data, instead of the interpolated data?

*Given our interest in comparing the drought conditions between different areas in our domain, meteorological series with a common length and with a common representativeness for each area are needed. The use of a gridded dataset allows us to have both these features, while the use of station data would present the problem of comparing series with different lengths and the problem of how to attribute the station data to certain portions of the territory. The possible effects of the interpolation method are discussed (with more details in Appendix A), in particular by focusing on the lack of elevation trend modelling in the interpolation method.*

- In section 2.3, according to the section title, the authors describe how they divide the areas based on elevation. However, from the text I understand that the division is actually based on ruggedness and not on elevation. Also, what is the difference between terrain roughness and ruggedness? Or are they the same? They seem to be used interchangeably. Please explain the difference and clearly state which one is used, or if they are the same, make sure to be consistent throughout the manuscript. Also, the authors state that they investigate orography, meaning the combination of elevation and ruggedness, but in the end they define groups based only on ruggedness, and not elevation. This should be corrected in the rest of the manuscript, where it is sometimes stated that the correlation between drought and orography is investigated.

*The title for Section 2.3 was chosen because the mean and standard deviation of the elevation (the latter representing the ruggedness of the terrain) are used to obtain two different classifications of the study area. To avoid confusion between the two metrics, as both are elevation-based, we have chosen to use the term "mean elevation", instead of just "elevation" when the first classification is mentioned in the rest of the manuscript. Furthermore, we acknowledge our error in using the term "roughness" instead of ruggedness in the manuscript—this typo is corrected in the revised manuscript. Finally, given that the results from both elevation-based classifications are contrasted throughout the manuscript, the term "orography" had been chosen to denote this type of study based on terrain characteristics, whether mean elevation or ruggedness is used; in order to avoid confusion, the term has been substituted for "terrain characteristics" throughout the revised manuscript.*

- The caption of figure 2 mentions ruggedness, while in the figure roughness is used. Also, in the caption "(d) correlation …" should be (f) and the caption states that it is the correlation between elevation and elevation standard deviation, while the axes in the figure describe mean elevation and terrain roughness. Although I understand that this is how the roughness is defined, it is better to be consistent and use the same term.

*Figure 2 has been changed according to Reviewer #2's comments.*

- From section 2.3 it seems that you are actually also investigating the differences in trends for different elevation groups and comparing that to the ruggedness groups. It would be good to make this more clear throughout the manuscript (e.g. also in the introduction). In addition, you could consider also showing the classification based on elevation in figure 2.

*A reference to the comparison between areas classified through mean elevation and terrain ruggedness has been added to the introduction: "Results obtained by focusing on either mean elevation or terrain ruggedness*

*are also compared, to understand if only elevation-related effect are present, or rather more complex interactions between meteorological drought and terrain characteristics." (Lines 57-59 of the revised manuscript). Furthermore, Section 2.3 has been thoroughly rewritten in order to address multiple comments by the Reviewers.*

• To calculate the parameters of the SPI, the authors use the maximum likelihood method and for the SPEI, they use probability weighted moments. Why not use the same method (if there is a good reason, please explain) and could this affect the results (e.g. the difference in the trends between SPI and SPEI)?

*The calculation for the two indices was based on the existing literature and the suggested methods for the distribution parameters estimation. For the SPI the maximum likelihood method is used as this is the method proposed in the literature (see https://doi.org/10.1007/s12145-014-0178-y, https://doi.org/10.1007/s11269-012-0026-0, both citing the formulas proposed in https://doi.org/10.1175/1520-0493(1958)086%3C0117:ANOTGD%3E2.0.CO;2). For the SPEI the proposed best method for parameter estimation is instead Hosking's PWMs. This method is proposed in Beguería et al., (2014), where the authors discuss briefly the effects of choosing one method over the other, stating that "[t]he SPEI series based on maximum likelihood were very similar to those based on the unbiased PWM method [...]. Given that calculation of the maximum likelihood estimation was about two-fold more time consuming, we conclude that the unbiased PWM method should be preferred for computation of SPEI series". Coherently, comparison between the results of the two method has been shown in our calculations to not have a meaningful impact on the results, as can be seen in the two following scatter plots between SPEI with ML and PWMs methods:*
**Scatter plot between SPEI-3 calculated with ML parameters and PWMs parameters**

[Figure]

***Scatter plot between SPEI-12 calculated with ML parameters and PWMs parameters***

[Figure]

*Although some slight differences are present, they are mainly located outside the 0 to –1 range used for drought definition in the study. Furthermore, all cells reported a Pearson correlation coefficient higher than 0.99 and RMSE lower than 0.09 between the series calculated with the two methods.*

• The section on trend analysis does not describe the method that is used for trend analysis, please add this. In addition, how are seasonal precipitation series defined and how are temperature series deseasonalised?

*The trend analysis method used for all series (precipitation, temperature and drought indices) is the one described in Section 2.4.3. Seasonal precipitation series are defined as the cumulative precipitation over the three month periods December-January-February (Winter), March-April-May (Spring), June-July-August (Summer), September-October-November (Fall). We have modified the manuscript in order to make this explicit (see Lines 175-177 of the revised manuscript). Deseasonalization was performed by applying the "deseason" function of the "Climate Data Toolbox for MATLAB" (https://doi.org/10.1029/2019GC008392), which calculates the seasonality as the mean of the detrended series for each month of the year. We acknowledge the missing citation for this function and the missing explanation of the method, and we have modified the manuscript accordingly (see Lines 177-179 of the revised manuscript).*

• The difference between drought runs and drought events is not very clear. Are drought runs based on one pixel and drought events based on multiple pixels? For the drought events, is the same method used as for the drought runs, but with the additional condition that 25% of the domain needs to be in drought? In addition, why did you choose 25% as threshold and what does "domain" mean? Is this the total case study area or the area within the different ruggedness areas? If the latter, could this introduce some bias in your results? Since the areas with low terrain ruggedness are very close together and the higher terrain ruggedness are more spread out, so less likely to be all in drought conditions at the same time?

*We acknowledge that the difference between the drought runs and events can be confusing, as (to our knowledge) no common way of referring to the two types of analysis is present in the literature. We called "drought runs" the droughts derived by analyzing the index series of a single pixel, derived via the application of thresholds and run analysis. "Drought events" (the name comes from https://link.springer.com/10.1007/s10584-022-03370-7) are instead droughts defined by considering all pixels in drought conditions (meaning all pixels that have a drought index value under a -1 threshold): if at least 25% of the pixels area experiencing drought conditions, a drought event is detected. This choice of area threshold is done to maintain consistency with the papers where this method was proposed, cited at Line 209-210 of the revised manuscript. In any case, following Reviewer #1's suggestion, we have decided to change the use of "drought run" and "drought event" in the manuscript to "local drought" (although still using the term run analysis as is common in the literature) and "region-wide drought event" to avoid confusion. Finally, the "domain" mentioned in the manuscript refers to the whole study area, not divided into different areas based either on mean elevation or on ruggedness.*

• The authors use a t-test to calculate the difference between the means of the two periods. Why was this method used for trend detection and not another method? What are the underlying assumptions of this method? Do they hold and what are the potential implications for the results? In addition, why not compare the number of drought events between the two periods (in addition to severity, duration)?

*The t-test method discussed in Section 2.4.6 is used to evaluate if a significant change in drought characteristics can be detected between the first and second half of the studied period. This is not an indication of the significance of the detected change; as stated at Lines 480-484 of the revised manuscript, this doesn't exclude that the changes between the two periods, even if significant, could be caused by the presence of particularly severe events not part of an overall trend. As the t-test is applied to compare the mean of a certain drought characteristic between two populations, the number of drought runs themselves could not be tested through this method. Trend detection was also performed on drought characteristics (see Line 367-372 of the revised manuscript), but almost no significant results could be obtained due to their discrete nature and the relatively small number of drought runs/events.*

• Figure 5 shows time series for a representative point in the domain, where is this point? And how can one point be representative if four different areas are investigated?

*Given that the drought indices series are not shown elsewhere, the aim of the figure was to give some context about differences between the SPI/SPEI at 3- and 12-month time scale (frequency of change, length of periods under the threshold…)—as such, the representativeness of the data was considered in regard to this aspect. To avoid confusion, and considering Reviewer #1 comments, we have decided to remove the figure.*

• From section 3.3, it seems that linear regression was used to calculate trends? This is not mentioned in the methods.

*The same trend detection procedure described in Section 2.4.3 is applied to the drought run data: given the lack of autocorrelation, this results in calculating the Mann-Kendall test and the Sen's slope.*

• When analysing the trends in drought runs and events, this seems to not be separated by area. Why not, since the main aim of the paper is to show the effect of ruggedness on drought trends?

*The relation between the change in drought run characteristics and ruggedness is studied by calculating the correlation between them. This was preferred to a comparison between the drought run characteristics' trends of areas defined through ruggedness values, given the low number of significant changes (as defined by the t-test) in drought characteristics. Furthermore, calculating drought runs from a drought index obtained from mean meteorological values belonging to an area instead of a pixel would have, in our opinion, created*

*confusion with the calculation of drought events. Finally, drought events are calculated on a region-wide level to have a point of comparison for the "local" drought runs' characteristics and their observed change.*

• There is no discussion of the results, only a summary. Are the results similar to the findings of the studies discussed in the introduction? And if not, why not? How are the results affected by the choice of methods?

*The Discussion section has been expanded by including a comparison between the presented study and similar ones found in the literature, indicating the novelty of our study and the originality of its results (see Lines 485-501 of the revised manuscript). In particular, we state that "[our] type of analysis is in common with a growing body of literature focused on the elevation effects on drought characteristics" (Line 488-489 of the revised manuscript), and that "our study shows that mean elevation, although certainly a variable to be considered, shouldn't be the only topographic variable taken into account" (Line 495-496 of the revised manuscript) given that "[i]n our analysis, using [a] different classification leads to stronger correlations between drought characteristics and topographical characteristics" (Line 499-500 of the revised manuscript).*

---

## Author Response (AR2)

**Authors' replies to the Reviewers of HESS-2023-218**

Dear Dr. Micha Werner,

Please find enclosed the revised manuscript entitled "60-years drought analysis of meteorological data in the western Po river Basin". First of all we would like to apologise for the delays in submitting this revised version. All the comments/suggestions raised by the reviewers, in *italic* below, have been addressed in the responses below and accordingly in the revised version. The changes have been highlighted in red and blue in the track-changed revised manuscript (using the track-changed LaTeX software).

We would like to take this opportunity to express our appreciation for the work of the Reviewers who provided very relevant and constructive comments and suggestions. We have addressed all the requests and amended the manuscript accordingly. We hope that the revised document is sufficient to warrant publication in HESS, and we look forward to hearing back from you.

**Editor**

*Dear Authors, we have received the referee reports on the revised manuscript. While both reviewers concede that the manuscript has improved in this revision, they also concur that the scope and scientific contribution of the manuscript is not clear. Several additional comments have also been raised. While I thank you again for your original revision, the issues raised on the scope are as yet not sufficiently addressed. In particular one of the reviewers provides some quite clear directions on how the scope could be clarified. I would therefore request you to consider these comments, and provide a clear response to the concerns raised.*

Indeed, the provided comments and suggestions led us to considerably reshape the manuscript in order to better clarify the scope of the research and the innovative aspects of the analyses. In summary, three major changes have been made to the manuscript (in order of significance):

1) The objectives of the work have been better declared in the introduction by stating three research questions, which are then answered in the concluding section. The methods and result sections are also related to the research questions thus improving the readibility and

clarity of the manuscript. A small change in the title has been made to indicate that the analyses are made on meteorological data.

2) Some of the analyses, for instance the preliminary analyses on precipitation and temperature data, have been removed. They were provided in previous versions of the paper for completeness but we agree that, not being directly related to the research questions, their presence would distract the reader from the focus of the work.

3) The limitations of the analyses, for instance regarding the temporal homogeneity of the used dataset and the limitations of detection studies respect to attribution ones, are now better declared in the text of the paper (see responses to the Reviewers).

**Reviewer #1**

*This is my second review of the manuscript "60-years drought analysis of meteorological data in the western Po river Basin".*

*The changes made the manuscript much clearer and easier to read. The methodology and the figures have been clarified and I believe the manuscript is now reproducible.*

*However, I still believe that the manuscript should aim for a clearer contribution to hydrology and earth system sciences. The conclusions have to be relevant for the hydrological/meteorological sciences in general rather than only for the current study area, which I don't think is the case here. For this, one idea would be for the manuscript to go beyond a simple correlation analysis (between trends and elevation) and analyze the physical link between those variables, that is, the causal association between them, to convince the reader that the conclusions found here can be extrapolated to other regions. There's a minor typo in the title, which should be "60-year drought analysis" rather than "60-years drought analysis."*

We acknowledge that the overall aim and contributions of the previous version of the manuscript were not conveyed appropriately and therefore the contribution to hydrology was unclear. We have therefore reworded the introduction and conclusions of the paper by stating three research questions which are then answered using evidence from the case study. The three questions are:

1) Are there temporal trends in drought indices such as SPI and SPEI, and how do these trends translate into changes in the characteristics of drought events, in terms of duration, severity, and intensity?

2) Is there a relationship between drought trends and topographical characteristics of the landscape? And if so, is elevation the topographical variable most correlated to these trends?

3) Do these conclusions change if drought events are defined at different spatial scales?

Even though the answer to these questions are given by referring to the particular case study, we believe that they may be of general interest and trigger further research to confirm or refute the generality of the findings. For instance, responding to question 1, we found that despite the worsening of drought conditions related to precipitation and temperature being clear, the effects on the characteristics of individual drought events are weaker. It would be of interest if this is the case also in other studies and why. In responding to question 2, we find that terrain ruggedness is better correlated to temporal drought dynamics than elevation, which has been proposed in other studies. Therefore mean elevation, although certainly a variable to be considered, shouldn't be the only topographic variable taken into account in drought change studies. We agree with Reviewer #1 that attributing this correlation to physical causes, e.g., change in atmospheric circulation behavior over the studied region, would have been even better but this should be done together with meteorologists and could be the objective of further research, beyond the scope of this paper. Finally, in responding to question 3, we find that drought characteristic changes at local and regional scales are different. While locally drought periods obtained from SPEI series show more pronounced increases in severity, duration and intensity than those obtained from SPI series, drought events at a region-wide scale show more marked shifts in severity and duration for SPI than for SPEI, denoting a more significant influence of regional precipitation patterns than of temperature on droughts at a regional scale. This is a non trivial result that could trigger further research in other regions. Also in this case the attribution to physical causes would have been better but, as stated above, beyond the scope of this paper.

In brief, even though the objective of attributing to physical causes the behavior obtained by our analysis, as suggested by Reviewer #1, is indeed the right final goal, we believe that the detection analysis done in our paper is anyway useful to the hydrologic community in raising interesting questions about the connection between different drought characteristics and their change at different spatial scales and in complex terrain settings. Analogous analyses in other regions of the world could provide information for attribution studies based on comparative hydrology (Falkenmark and Chapman, 1989, ISBN: 9231025716; Blöschl et al, 2013, https://doi.org/10.1017/CBO9781139235761).

**Reviewer #2**

*The overall aim and added value of the paper is still unclear. It seems that the main novelty is the analysis of drought characteristics in relation to terrain ruggedness. However, a large part of the methods and results discusses analyses that don't seem to contribute to this aim. The parts of the analysis that do contribute to this aim are a bit hidden in between the other analyses.*

Reviewer #2 is right. Our revision of the first version of the manuscript has biased the discussion toward one of the objectives of the paper, i.e., the correlation analysis between drought (change) characteristics and terrain ruggedness, thus resulting in an unclear scope. In this version, we state more clearly the objectives of the paper by stating three research questions in the introduction section that should help the reader understand why certain analyses have been done. The three questions are:

1) Are there temporal trends in drought indices such as SPI and SPEI, and how do these trends translate into changes in the characteristics of drought events, in terms of duration, severity, and intensity?
2) Is there a relationship between drought trends and topographical characteristics of the landscape? And if so, is elevation the topographical variable most correlated to these trends?
3) Do these conclusions change if drought events are defined at different spatial scales?

The Discussion section in the paper summarizes now the answers to these questions.

*While the changes related to terminology make things more clear, the results are still rather difficult to follow and interpret. The paper reads more like a subsequent application of many different statistical analyses rather than a coherent methodology to investigate the differences across terrain ruggedness. It seems that many of the reported results could be moved to the supplementary material, since they don't contribute to the overall aim and don't show significant results. In addition, throughout the results elevation and terrain ruggedness are reported intermittently and are sometimes mixed up. Sometimes they are compared to each other and in other cases not, without an apparent reason. The results present several different versions of mean drought characteristics, sometimes referring to a mean across all pixels and sometimes referring to a mean value across different drought runs in one pixel. This is very confusing and it is different to keep track of what kind of mean a specific section is talking about.*

We hope that the rewriting of introduction and conclusions, plus the clarifications in the other sections of the paper allow now an easier readability of the paper. As said above, the research questions asked are more than one and the analyses made are instrumental in tackling them. Nevertheless, we have removed some of the analyses which are not explicitly related to the research questions, such as the analyses on precipitation and temperatures that were done before calculating the drought indices.

Regarding the analyses on the connection between elevation and ruggedness and droughts, we acknowledge that the previous version of the paper was sometimes confusing and inconsistent because the results obtained stratifying drought (change) characteristics based on the two variables were not always reported, giving more space to the ruggedness. In the new version of the paper, we show all analyses for both elevation and ruggedness and we show that, although both are correlated to drought characteristics and their change in time, terrain ruggedness is a better predictor than mean elevation, and therefore a potentially useful variable to be considered in other drought change studies.

*Overall, I do believe that the paper could make a valuable contribution. However, the presentation still needs to be improved. It would be good if the authors present an overall methodology, detailing why they are doing each of the different tests and analysis and how this contributes to the overall research aims (which seems to be investigating the influence of elevation terrain ruggedness). Similarly the results could be presented in a more coherent way, with a focus on the overall research aims.*

We thank a lot Reviewer #2 for this suggestion that we have tried to follow in the best way we could. Indeed the previous version of the manuscript had relevant clarity issues that, we believe, have been resolved with the revision made.

*Some specific remarks:*

*The added value of the regional drought analysis is not clear to me. There don't seem to be any interesting results related to this analysis and this analysis does not contribute to the aim of the paper which is to look at differences in changes in drought across differences in terrain ruggedness.*

The relevance of the regional drought analysis is explained better in the revised manuscript. One of the research questions (the third one) is dedicated to whether different results may be obtained by conducting drought analyses at the local and regional scales. Interestingly, drought characteristic changes at local and regional scales are different. While locally drought periods obtained from SPEI series show more pronounced increases in severity, duration and intensity than those obtained from SPI series, drought events at a region-wide scale show more marked shifts in severity and duration for SPI than for SPEI,

denoting a more significant influence of regional precipitation patterns than of temperature on droughts at a regional scale. This is a non trivial result that could trigger further research in other regions.

*Lines 50-54: "On the other hand, studies considering temperature values have consistently shown rising temperatures, and thus a rise in evaporative demand, to be a main factor in drought increase, even when significant changes in precipitation patterns were detected." Which studies? This needs references.*

Reviewer #2 refers to lines 35-37 of the manuscript after the first round of revisions. The sentence refers to those studies cited in the previous sentences that have also considered temperatures, and therefore the SPEI, in detecting drought trends. We have rephrased the sentence to "Among these studies, those also considering temperature values consistently showed rising temperatures, and thus a rise in evaporative demand, to be a main factor in drought increase."

*Methods: I still don't see the added value of using the gridded data and analysing 227 grid points instead of analysing the station data at the 200 stations. The authors mention that it is difficult to attribute stations to the different regions, but it seems that you would know the elevation of each station, so it should be possible to assign the stations to a region? If instead the problem is that many of these stations have a time series that is too short, this should be highlighted in the manuscript more clearly and the implications for the results of this study should be discussed. Then the statement that the dataset is based on 200 stations is not entirely accurate and this could affect the results of your analysis. If the interpolated dataset is based on only 25 stations or a bit more for the period before 1990 and only after 1990 the dataset increases to 371 (from the manuscript or appendix it is not clear how many stations were available in each year exactly) then how will this affect the comparison of the drought event characteristics before and after 1990? Similarly, when evaluating the differences in drought and drought characteristics across the region, this may be heavily influenced by the stations that were available for the interpolation in each particular year. In the earlier years, with less stations you would probably expect variability to be lower than when you have more stations available for the interpolation. It seems like you are hiding some of the limitations of the dataset by using the interpolated dataset and not considering the effects it may have on your analyses.*

This is indeed a relevant comment that should be discussed in the paper. Information on the temporal evolution of the number of stations is available at the url https://www.arpa.piemonte.it/scheda-informativa/spazializzazione-dei-dati-temperatura-precipitazione-griglia. Regarding precipitation the number of stations has been relatively high throughout the whole period. Regarding temperature indeed the number of stations is

low before 1990, but their information has been added to ERA40 data as a background information. We have accepted a compromise by using this official database for the region, which is more detailed and accurate than any other gridded database of precipitation and temperatures available in the area (and for larger areas). The advantage of using this database is the availability of spatially consistent information for a long time period (1950s to 2020s). The drawback is that being an interpolated product, the change in the density of the ground stations, which has been significant in the years, may have had an effect on the results, particularly on extremes occurring locally. We now discuss the issue in the revised manuscript. Nevertheless we believe that the advantages of having a long-term database are superior to the disadvantages due to its potential lack of homogeneity. We couldn't conduct a sensitivity analysis because we do not have access to the station data used to produce the gridded product (we can download the station data only for the most recent period, 2000s-2010s). Appendix A provided the information we could retrieve about the construction of the database. Being work done by others, we prefer in the new version of the paper to remove Appendix A and referring to the original documents in the text.

*In the methods, line 146-147, the authors mention dividing the area in four groups based on both elevation and terrain ruggedness. However, it seems the groups are based only on terrain ruggedness? The four groups based on elevation are not reported anywhere.*

Reviewer #2 is right (even though the lines are 125-127). The figure just reports the classification by terrain ruggedness, while also showing the corresponding mean elevation.  We have rephrased the sentence and the figure caption to avoid confusion. The sentence now reads "The landscape is classified in areas with similar topography. Four distinct areas of an almost equal number of cells are identified based on terrain ruggedness, which represent the plains, the hilly region, and the lower and higher mountains respectively. Figure B1 shows the classified areas and the fact that mean elevation and terrain ruggedness are highly correlated. However, the advantage of using terrain ruggedness over mean elevation is that, in our study area, the hills in the center-south of the region are distinguished from the eastern flat part of the region, despite having similar mean elevation."

*The description of the trend analysis in 2.4.3 is still not clear. Here you are describing the steps for pre-whitening methods mainly. Please explain the trend analysis (e.g. Sen's slope and Mann-Kendall test) in more detail, this may not be trivial for every reader.*

A brief explanation of the Sen's slope and Mann-Kendall test has been added to the text. The following sentence has been added: "The trends are estimated using the Theil-Sen slope estimator (Theil, 1950; Sen, 1968), i.e., by calculating the median slope between the

indices values for all possible month pairs. The significance test is performed through the Mann-Kendall test (Mann, 1945), which is a non-parametric (distribution-free) alternative to the linear regression slope test available in regression analysis. To improve the power of the test, deseasonalization and pre-whitening of the data are performed."

*Methods section 2.4.4: Although you have changed the naming to a local drought analysis, you may still want to indicate at the start of the paragraph, that this analysis is performed pixel by pixel.*

Done, thank you.

*Methods section 2.4.5. From this section, it is still not clear to me whether by region you mean the entire study area or if you analyse "local regional" droughts, i.e. local areas of multiple cells that are experiencing drought. It does become clear later on, but should also be mentioned here.*

Thank you, we rephrased the sentence as "In contrast to local droughts, which are calculated from a series of index values belonging to one cell, region-wide drought events are evaluated by considering what happens in the entire region." Then the procedure is explained in detail.

*Line 269-270: "used in the cited papers", this is not clear, please just cite the proper references here, e.g. used by Name (year).*

Done, thank you.

*Results 3.1, at the end of the paragraph, lines 340-344, the authors state that the results agree with the results from other studies with the same data set. So why are the authors redoing this analysis? The added value is not clear to me.*

The precipitation and temperature trend analyses were done for completeness but have now been removed from the paper since not directly related to the three research questions stated in the introduction. The sentence has therefore disappeared from the revised manuscript.

*Results 3.2, line 374-376. This line refers to Figure 6 which shows the division according to terrain ruggedness, yet this line mentions a difference between altitudes. Should this be terrain ruggedness?*

We guess Reviewer #2 refers to lines 308-309 (not sure why we have a different numbering) where the sentence was "...despite the trend in annual precipitation being not significant and the temperature trends having a lower slope coefficient than at higher altitudes (Figure 6)". Indeed the sentence was unclear. The revised manuscript, heavily revised in this part,

does not include the sentence anymore. Figure 6 has been removed and only part of it is now included in the new Figure 3.

*Figure 6 shows trend analyses for the different groups, but this analysis is not described in the methods section (i.e. how are you calculating the group mean prec, max T, min T, etc.?)*

Figure 6 has been removed and only part of it is now included in the new Figure 3. The classification by terrain ruggedness of the indices is explained in the Figure caption by the sentence "Trend analysis on drought indices calculated from data belonging to areas defined by terrain ruggedness inside cells. The colour of the circles represents the slope coefficient of the trend, while the inner radius of the circles represents the significance of the trend (a smaller inner radius represents a more significant trend). The black circles denote a significance level of 5%."

*Results 3.3.1 Here correlations with elevation are discussed and the correlation with terrain ruggedness only briefly mentioned, even though the initial aim of the paper is to show the importance of including terrain ruggedness instead of elevation.*

The correlation of local drought characteristics with mean elevation and terrain ruggedness is not very different. Terrain ruggedness seems to be more significantly correlated with runs characteristics evaluated with indices applied for the long (12 month) duration. For the short (3 months) duration perhaps mean elevation is better (see new Table 1). For changes in run characteristics, instead, the correlation with terrain ruggedness is always superior to the correlation with mean elevation (see new Table 2).

*Figure 7 is discussed in section 3.3.2, after the discussion of tables 1 and 2, but the figure comes before the tables, this is quite confusing.*

Right. Now Figure 4 (what was Figure 7) comes first and the tables next. LaTeX is to blame :-)

*The caption of figure 7 is a bit confusing. All four subfigures seem to show decreases and increases (with a downward or upward arrow), but the caption says a and b show a decrease and c and d an increase? Also, the caption would be more clear if the sub figures are discussed in alphabetic order (instead of a, c, b, d).*

We agree, the caption has been reworded.

---

## Author Response (AR3)

**Authors' replies to the Reviewer of HESS-2023-218**

Dear Dr. Micha Werner,

Please find enclosed the revised manuscript entitled "60-years drought analysis of meteorological data in the western Po river Basin". We thank you for your comments, which have been very helpful to better the quality of our work. In particular, we have made many changes to the text improving the clarity of writing and we have partly changed the figures insuring more consistency between colour scales (as in Figure 3 of the revised manuscript) and greater differentiation between ranges of values (as in Figure 4 of the revised manuscript). We have addressed all your suggestions (as seen in the red/blue highlights in the track-change file provided), and we would like to offer more complete responses to some of them, in *italic* below.

*"Even though the change is less significant that the one obtained by analysing the indices themselves.". It is not so clear what is meant with this sentence.*

The sentence was meant to convey the difference between the results obtained from studying trends in the drought indexes' series (SPI/SPEI) and in the drought run characteristics calculated from those series. The manuscript has been changed for better clarity.

*Mention is made of drought impacts being diversified. Whilst I agree this is the case, the discussion in this section is primarily on the hazard dimension of drought risk, and not on the consequence (impact) dimension. So, the diversification of drought impacts does not follow from the discussion. I would suggest to rephrase this such that the focus remains on dimensions of drought hazard.*

We have changed the manuscript as to not mention impacts, but rather "drought characteristics" and "the frequency of occurrence of drought periods of a certain magnitude" (see Line 68-69 of the revised manuscript).

*There is discussion that a threshold value of -1 is used to identify drought events, and then in Figure 2 it is mentioned that this is not used in this study. This is somewhat confusing. Please clarify. It may be useful to be clearer on how a drought event is identified. As I understand it, if one or more months have an index value of below -1, then all preceding and following months are considered the same event, until a positive (or zero) index value is reached.*

We have rephrased the explanation of the local drought period definition, citing immediately the inclusion of the onset/offset (i.e. months with negative index values preceding and following the months under the threshold, see Lines 224-227 of the revised manuscript). Furthermore, we have changed Figure 2, excluding the panel showing the method not used in the paper as to limit redundancy, and we have instead added a panel to better explain the region-wide drought event definition (see the response to the next comment).

*I assume that each cell is first evaluated if it is in a condition of drought (i.e. the local drought is evaluated first), before the regional analysis is done. This means that mention of the condition being below -1 is not quite correct as in the previous it was described how drought conditions were identified at the local level, which can include some values between 0 and -1.*

Our definition of region-wide drought events follows that of the cited paper at Lines 245 of the revised manuscript, which uses a simple -1 threshold in order to define cells in drought conditions (other citations present in the previous manuscripts were omitted, in order to be more concise as they were both based on González-Hidalgo et al., 2018). The choice to maintain this procedure was done for consistency with the cited paper. We have made this different mode explicit, see Lines 245-8. Furthermore, the choice to use a temporal aggregation through the inclusion of months with some cells in drought conditions before/after the months with more than 25% of the cells in drought conditions is done to mimic the aggregation done by González-Hidalgo et al., 2018. The aggregation through this analytic method can be easily applied in our case due to the limited study area.

*While I understand that a regional drought event is considered to persist when less than 25% of the area, similar to the temporal persistence for the local analysis, it would appear to me that the spatial correlation of the areas is relevant. Would this approach not inflate the length of drought periods spatially for regional areas that have a higher climatic variability, such as the more rugged regions?*

We thank for this very relevant comment and the opportunity to address it. While we acknowledge the possibility of overextending the length of region-wide drought events for the areas with a higher climatic variability, we did not find this happening in our case. For example, see the two attached figures below, showing the percentage of time in which each cell has been part of region-wide drought events. In general, it appears that the alpine chain is not more likely to be part of the events compared to other areas, even in cases where events span the whole region.

[Figure]

*Figure 1: Percentage of time in which each cell is part of region-wide drought events calculated through SPI-12. The numbers refer to the events shown in Figure A3 of the revised manuscript.*

[Figure]

*Figure 2: Percentage of time in which each cell is part of region-wide drought events calculated through SPEI-12. The numbers refer to the events shown in Figure A3 of the revised manuscript.*

*L328-330: Note that some of these results are trivial. There are by definition less frequent drought events when comparing SPI-3 and SPI-12, or SPEI-3 and SPEI-12 as the longer averaging window smooths the signal, resulting in attenuation and pooling.*

*That drought events at the longer time scale are longer though less frequent is again somewhat trivial as it is inherent to the method.*

We agree with the comment and have decided to keep these observations on the data but making explicit that this result is to be expected (see Line 322 of the revised manuscript as well as Line 398). Furthermore, we have changed Figure 4 in order to make it more readable, using two different colour ramps for 3 and 12 month drought characteristics.

*Figure 5: This figure is somewhat confusing. In the text it is noted that there is an on average increase or decrease depending on the indicator. This means that there are cells that also show the opposite trend of that indicator. But the figure shows only where there is an increase (upper panels) or a decrease (lower panels). Should this not also display those cells (using the appropriate symbol?). Of perhaps clarify in the text that only the one direction is shown.*

*L372: Mention is made of a higher number of shorter duration droughts are found in the alpine chain. But this cannot be seen in the figure (see comment above) as it shows only increase in length. This is somewhat confusing as the figure does not corroborate the text.*

We have decided to overhaul Figure 5 in order to make it more readable and clearer, representing both types of changes (higher/lower severity and longer/shorter duration) in each panel and dividing between drought characteristic and drought index studied. Furthermore, the mean cited in the text of the Figure refers to the mean over the first and the second period in which the series was divided. The text has been amended as to make this clearer.